

# Does feedback temperature dependence influence the slow mode of the climate response?

Tim  Rohrschneider[1,2], Jonah Bloch-Johnson[3], and Maria Rugenstein[4]

[1]Max Planck Institute for Meteorology, Hamburg, Germany
[2]International Max Planck Research School on Earth System Modelling, Hamburg, Germany
[3]University of Reading, Reading, UK
[4]Colorado State University, Colorado, US

**Correspondence:** Tim Rohrschneider (tim.rohrschneider@mpimet.mpg.de)

**Abstract.** Atmosphere-Ocean General Circulation models (AOGCMs) are a necessary tool to understand climate dynamics on centennial timescales for which observations are scarce. We explore to which degree the temperature dependence of the climate radiative feedback influences the slow mode of the surface temperature response. We question whether long-term climate change is described by a single e-folding mode with a constant timescale which is commonly assumed to be independent of temperature or forcing and the evolution of time. To do so, we analyze AOGCM simulations which have an integration time of 1000 years and are forced by atmospheric $CO_2$ concentrations ranging from 2 times (2X) to 8 times (8X) the preindustrial level. Our findings suggest that feedback temperature dependence strongly influences the equilibrium temperature response and adjustment timescale of the slow mode. The magnitude and timescale of the slow mode is approximately reproduced by a zero-dimensional energy balance model that has a constant effective heat capacity and incorporates a background feedback parameter and a coefficient for feedback temperature dependence. However, the effective heat capacity of the slow mode increases over time, which makes the adjustment timescale also time-dependent. The time-varying adjustment timescale can be approximated by a multiple timescale structure of the slow temperature response, or vice versa, a multiple timescale structure of the slow temperature response is described by a time-dependent timescale. The state-dependence and time-dependence of the adjustment timescale of long-term climate change puts into question common eigenmode decomposition with a fast and a slow timescale in the sense that the slow mode is not well described by a single linear e-folding mode with a constant timescale. We find that such an eigenmode decomposition is valid at a certain forcing level only, and an additional mode or a multiple mode and timescale structure of the slow adjustment is necessary to reproduce the details of AOGCM simulated long-term climate change.



## 1 Introduction

Long-term climate change is determined by the slow mode of the surface temperature response. We analyze the slow mode in
light of temperature-dependent radiative feedback and time-varying adjustment timescales using abrupt $CO_2$ experiments with
Atmosphere-Ocean General Circulation models (AOGCMs). The slow mode approximately describes the temporal tempera-
ture adjustment in response to radiative forcing on a multi-centennial timescale. In this study feedback temperature dependence
describes how the radiative feedback of the climate system depends on the global mean surface temperature change. As a result,

higher forcing will cause a greater change in feedback, as it will cause a greater increase in temperature. Our study is motivated
scientifically by understanding and predicting long-term climate change beyond year 2100.

Using proxies or complex climate models, studies on paleo climates demonstrate that the climate feedback depends on the
climate state (e.g. Kutzbach et al., 2013; Schaffer et al., 2016; Wolf et al., 2018; Farnsworth et al., 2019). Recent research on

modern-day boundary conditions demonstrates that feedback temperature dependence of the global feedback becomes impor-
tant in the case of high forcing (e.g. Roe and Baker, 2007; Roe and Armour, 2011; Meraner et al., 2013; Bloch-Johnson et al.,
2015; Rohrschneider et al., 2019). The latter studies are based on Charney-type feedbacks (Charney et al., 1979) and mostly
focus on the equilibrium surface temperature change in response to forcing input such as a quadrupling of the atmospheric
$CO_2$ concentration above preindustrial levels. The equilibrium temperature change in response to radiative perturbations can

be described by a zero-dimensional energy balance model that incorporates a parameter for temperature-dependent feedback
(Zaliapin and Ghil, 2010; Bloch-Johnson et al., 2015). The zero-dimensional model describes the relationship between the
global mean radiative forcing input $F$ and global mean equilibrium surface temperature perturbation $T(\infty)$,

$$-F = (\lambda_\mathrm{b} + aT(\infty))T(\infty), \tag{1}$$

where $\lambda_\mathrm{b}$ is the initial or background feedback parameter and $a$ is the coefficient for feedback temperature dependence. Ac-

cording to this framework, the equilibrium temperature response $T(\infty)$ depends nonlinearly on the radiative forcing $F$. By
contrast, it scales linearly with forcing in the case of zero feedback temperature dependence. Furthermore, feedback tempera-
ture dependence does not only influence the equilibrium temperature change but also the temporal behavior of the temperature
adjustment. The temporal adjustment of this zero-dimensional energy balance model is described by

$$C\frac{\mathrm{d}T}{\mathrm{d}t} = N \text{ or } C\frac{\mathrm{d}T}{\mathrm{d}t} = F + (\lambda_\mathrm{b} + aT)T, \tag{2}$$

where $N$ is the net top-of-the-atmosphere (TOA) radiative imbalance compared to steady state, and $C$ is the effective heat ca-
pacity of the global system. The latter is an effective quantity because it depends on the ocean circulation and does not directly
represent the ocean mass. Integrating analytically the right-hand side of Eq. (2) gives a timescale that depends on the strength
of the forcing because the feedback changes with warming. The state-dependence of the temporal adjustment of the climate
response has been demonstrated conceptually and with simulations of a single AOGCM (Rohrschneider et al., 2019).






The global mean surface temperature response of the climate system is approximately described by a fast mode that acts on a decadal timescale and a slow mode that acts on a multi-centennial timescale. In general, the two-timescale approach has been found a good approximation for complex model behavior (e.g. Held et al., 2010; Winton et al., 2010; Geoffroy et al., 2013a, b; Rohrschneider et al., 2019). Recent studies suggest that the slow mode is either a function of the Earth's deep ocean
component (e.g. Held et al., 2010; Winton et al., 2010; Geoffroy et al., 2013a, b), or associated with a radiative feedback (e.g. Armour et al., 2013; Proistosescu and Huybers, 2017). Mathematically, these concepts are equivalent (Rohrschneider et al., 2019). Commonly, the concept of a fast mode and a slow mode is based on eigenmode decomposition. Using linear eigenmode decompostion, the temperature evolution $T(t)$ over time is approximated by multiple exponential modes ($n$),

$$T(t) = \sum T_n(\infty)(1 - e^{-t/\tau_n}), \tag{3}$$

with $T_n(\infty)$ the amplitude, and $\tau_n$ the e-folding timescale. The question arises whether the two-timescale approach with a slow e-folding mode and a constant timescale is still an appropriate description of long-term climate change. According to theory, feedback temperature dependence makes the adjustment timescale continuous and makes it depend on the forcing. Furthermore, the thermal inertia of the slow mode, mathematically described by the effective heat capacity, can change over time or with climate state such that the influence of feedback temperature dependence on the temporal adjustment is modified. For
instance, the ocean circulation may change and the heat flux into the deep ocean may become less efficient (Rugenstein et al., 2016b), which would cause $C$ to increase.

It is important to know the mode and timescale structure of the temperature response in order to make accurate predictions and understand the temporal temperature adjustment at different timescales even without having an underlying physical model.
It is debated how many adjustment modes exist and are necessary to reproduce the complex system response (Olivie et al., 2012; Caldeira and Myhrvold, 2013; Knutti and Rugenstein, 2015; Proistosescu and Huybers, 2017). We question whether the slow mode is a single eigenmode with a single constant timescale in the common sense. The timescale is commonly assumed to be independent of temperature or forcing and the evolution of time.

We use abrupt $CO_2$ experiments with multiple AOGCMs in order to answer this research question. Using multiple AOGCMs with different radiative responses and atmospheric and oceanic parameterizations allows us to assess whether the influence of feedback temperature dependence on long-term climate change is substantial. The AOGCM experiments used here are the only publicly available experiments to date that have an integration time of at least 1000 years and provide three different forcing levels (Rugenstein et al., 2019). Thus, we can explore the changes in the slow adjustment with forcing and analyze long-term
climate change on a multi-centennial timescale. We compare the low-end and high-end forcing range considered in CMIP6 scenarios (O'Neill et al., 2016) to have a large signal; that is, two times (2X) and eight times (8X) the preindustrial $CO_2$. The $CO_2$ concentration is held constant throughout the simulation time so that we can explore the underlying dynamics. In mathematical terms, the radiative forcing is a step function input.





Section 2 provides conceptual insights on the slow mode. Section 3 outlines the experimental strategy and characterizes the AOGCM properties in simulating the climate response. In section 4, we analyze the equilibrium response and timescale of the slow mode in light of temperature-dependent feedback. In section 5, we analyze the interplay between state-varying and time-varying adjustment timescales and demonstrate the limits of eigenmode decomposition in terms of the two-timescale approach.

## 2 Conceptual insights

Before exploring the slow mode's behavior in AOGCMs, we provide conceptual insights about the slow mode using simple climate models. These simple models are energy balance models and outlined in detail in Geoffroy et al. (2013a), Geoffroy et al. (2013b), Armour et al. (2013), Rohrschneider et al. (2019), among others. We bring together these existing concepts to lay out the parameter dependencies of the slow mode in order to provide a solid basis and motivation for our experimental analysis.

A way to represent the global mean surface temperature response to forcing is to assume two effective regions, $T = (\chi - 1)T_\mathrm{F} + \chi T_\mathrm{S}$, where $\chi$ is the effective fractional area:

$$C_\mathrm{F} \frac{\mathrm{d}T_\mathrm{F}}{\mathrm{d}t} = F + (\lambda_\mathrm{F} + a_\mathrm{F} T_\mathrm{F}) T_\mathrm{F}. \tag{4}$$

and

$$C_\mathrm{S} \frac{\mathrm{d}T_\mathrm{S}}{\mathrm{d}t} = F + (\lambda_\mathrm{S} + a_\mathrm{S} T_\mathrm{S}) T_\mathrm{S}. \tag{5}$$

$F$ is the radiative forcing, $C$ is the constant effective heat capacity, $\lambda$ is the background feedback parameter, $a$ is the coefficient for feedback temperature dependence. Each region behaves similarly to Eq. (2), and according to this framework, the climate response is characterized by a fast mode $T_\mathrm{F}$ and a slow mode $T_\mathrm{S}$. In this paper, we analyze the influence of feedback temperature dependence on the slow mode only. Positive feedback temperature dependence causes the equilibrium response of the slow mode to increase. Furthermore, feedback temperature dependence introduces a timescale that depends on the strength of the forcing. Considering the temporal behavior, the thermal inertia of the slow mode is represented by a single heat capacity which is much higher than the heat capacity of the fast mode ($C_\mathrm{F} \ll C_\mathrm{S}$). At this point, $C_\mathrm{S}$ is constant over time and does not change with the climate state. The slow mode of the surface response is thought to be coupled to the state of the deep ocean or being an effective region.

Another conceptual framework with a fast mode $T_\mathrm{F}$ and a slow mode $T_\mathrm{S}$ is the two-layer ocean model with ocean heat uptake efficacy and feedback temperature dependence (Held et al., 2010; Winton et al., 2010). This model combines time-dependent feedback due to the evolution of two different state-variables and state-dependent feedback due to temperature-dependent feedback. The model configuration with ocean heat uptake efficacy and feedback temperature dependence is given by

$$C \frac{\mathrm{d}T}{\mathrm{d}t} = F + (\lambda_\mathrm{b} + aT)T - \epsilon \eta (T - T_\mathrm{D}) \tag{6}$$





$$C_{\mathrm{D}}\frac{\mathrm{d}T_{\mathrm{D}}}{\mathrm{d}t} = \eta(T - T_{\mathrm{D}}) \tag{7}$$

where $C \ll C_{\mathrm{D}}$ are the heat capacities of the upper- and deep-ocean, $\lambda_{\mathrm{b}}$ is the background feedback parameter and $a$ the coefficient for feedback temperature dependence. The parameter $\eta$ is the heat transport efficiency and $\epsilon$ the efficacy factor for ocean heat uptake. The slow component is approximated by

$$T_s(t) \approx \frac{\sqrt{\Lambda^2 - 4aF} - \sqrt{\Lambda^2 - 4aF - 4a\epsilon\eta T_{\mathrm{D}}(t)}}{2a} \quad \text{with } \Lambda = \lambda_{\mathrm{b}} - \epsilon\eta \tag{8}$$

after the fast contribution from the surface, as derived in (Rohrschneider et al., 2019). Following this conceptual framework, the slow mode is a function of the deep ocean component $T_{\mathrm{D}}$ because the slow mode emerges from the heat transport into the deep ocean and the convergence of the state-variables over time towards the same equilibrium temperature perturbation.

Using linear model versions without feedback temperature dependence, the two-region model and the two-layer model are mathematically equivalent. Although no analytical solution of the coupled two-layer model with feedback temperature dependence exists to date, we can approximate the temperature and radiative response associated with the slow mode by a single effective region (Eq. 5), having a single heat capacity. However, the parameters of the two-layer model modify the inertia of the slow mode. For instance, the parameter for the efficiency of ocean heat uptake $\eta$ is an inertia parameter, and changes in ocean heat uptake cause $C_{\mathrm{S}}$ to increase or decrease. Commonly, we assume that the parameters which describe these simple models are constant. In that respect, we emphasize that the slow mode's response is described by

$$C_{\mathrm{S}}\frac{\mathrm{d}T_{\mathrm{S}}}{\mathrm{d}t} = N_{\mathrm{S}} \tag{9}$$

where $N_{\mathrm{S}}$ is the TOA imbalance associated with the slow mode. After having explored the imprint of feedback temperature dependence on the slow mode, we analyze the interplay of state-varying and time-varying adjustment timescales. The former arises from the presence of feedback temperature dependence while the latter arises from the inconstancy of $C_{\mathrm{S}}$ according to Eq. (5,9).

As a starting point, we analyze the parameter dependencies of the equilibrium response and timescale of the slow mode $T_{\mathrm{S}}$ (Fig. 1). For illustration only we use the more complicated two-layer ocean model (Eq. 6,7) and focus on the temperature dependence of the global feedback and additionally on the efficiency of ocean heat uptake. The latter allows us to illustrate the changes in the slow temperature adjustment which emerge from the changes in ocean heat uptake. We show the characteristic temperature adjustment of the slow mode $T_{\mathrm{S}}$ assuming positive feedback temperature dependence (Fig. 1a). The temperature response increases relatively to the forcing level in the case of positive feedback temperature dependence and decreases relatively to the forcing level in the case of negative feedback temperature dependence.

Fig. 1b shows the parameter dependencies of the steady temperature response a of the slow mode. The magnitude of $T_{\mathrm{S}}(\infty)$ increases with more positive feedback temperature dependence $a$, with higher forcing leading to a more nonlinear response.

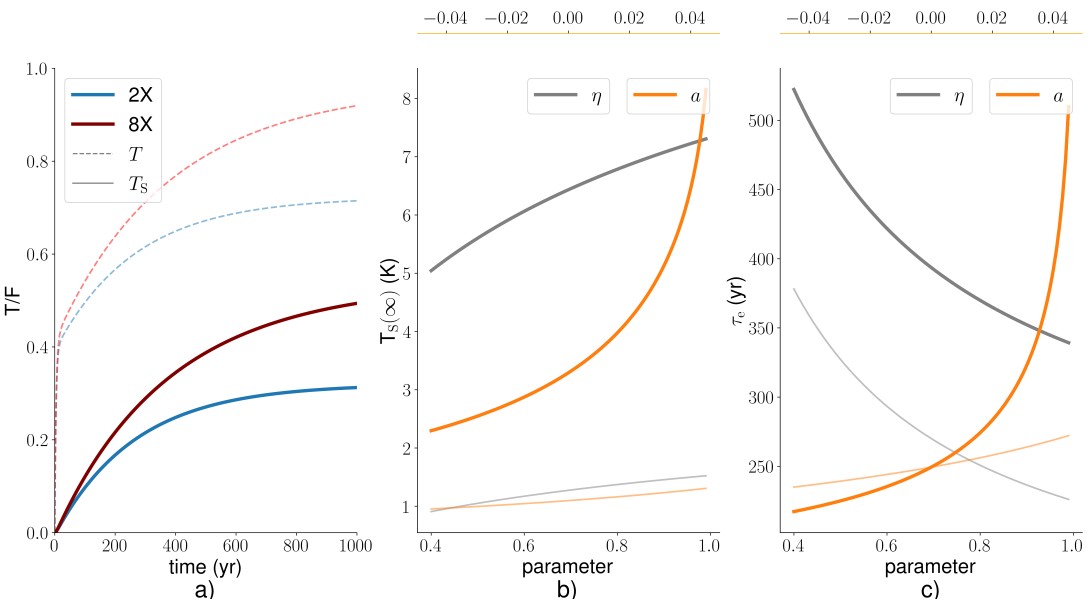

**Figure 1.** a) The slow mode $T_S(t)$ (solid) in the two-layer ocean model normalized by the radiative forcing $F$ in an idealized case. We further show the surface temperature response including the fast mode (dashed). b) The parameter dependencies of the equilibrium temperature response $T_S(\infty)$, and c) the approximated e-folding timescale timescale $\tau_e$ (Eq. 10) of the slow mode. As a reference, we use $F = 4 \text{ W m}^{-2}$ (2X) (thin transparent in b) and c) ) and $F = 12 \text{ W m}^{-2}$ (8X) (opaque in b) and c) ). The reference parameters are $a = 0.04 \text{ W m}^{-2} \text{ K}^{-2}$, $\epsilon = 1.5$, $\eta = 0.7 \text{ W m}^{-2} \text{ K}^{-1}$, $\lambda_b = 1.5 \text{ W m}^{-2} \text{ K}^{-2}$, $C = 10 \text{ W m}^{-2} \text{ K}^{-1} \text{ yr}$, $C_D = 100 \text{ W m}^{-2} \text{ K}^{-1} \text{ yr}$.

We further show the parameter dependence of the slow mode $T_S$ on the efficiency $\eta$. Considering the efficiency of ocean heat

uptake $\eta$, more efficient heat transport does not change the climate sensitivity $T(\infty)$ but does change the equilibrium response of the slow mode $T_S(\infty)$. The higher $\eta$ is, the lower is the magnitude of the fast mode $T_F(\infty)$ and the stronger is the magnitude of the slow mode $T_S(\infty)$. We neglect the efficacy factor $\epsilon$ in Fig. 1. The efficacy factor $\epsilon$ mimics the pattern effect, describing the time-dependence of the radiative feedback on the pattern of surface warming, (Winton et al., 2010; Stevens et al., 2016) and thus represents the differences between regional feedbacks. The higher the efficacy factor for ocean heat uptake $\epsilon$, the less

efficient is the radiative response associated with the slow mode and the higher is the equilibrium response of the latter.

Fig. 1c shows the parameter dependencies of the timescale of the slow mode ($\tau_e$). For the sake of simplicity, we assume that the evolution of the slow adjustment is approximately described by a single e-folding mode,

$$T_S(t) \approx T_s(\infty)(1 - e^{-t/\tau_e}). \tag{10}$$

This makes the strong assumption that the slow mode adjusts on a single timescale that presumably depends on the strength of the forcing. For instance, the analytical solution of the slow mode in the two-region model (Eq. 5) is more complex than this equation (Rohrschneider et al., 2019), and the deviations between these expressions can be interpreted as a time-varying,





effective timescale $\tau(t)$. Focusing on the temperature dependence of the global feedback, the timescale of the slow mode $\tau_e$ increases with more positive feedback temperature dependence, $a$. As with the equilibrium response $T_S(\infty)$, the timescale of

the slow mode increases far more with higher forcing $F$. By contrast, more efficient heat transport into the deep ocean ($\eta$) decreases the timescale of the slow mode $\tau_e$, so that the equilibrium response is reached at an earlier time.

Using these concepts, we now analyze the outcome of the AOGCM experiments. We focus on the slow mode in light of feedback temperature dependence and the two-region model as the underlying framework. We denote Eq. (5) as the zero-

dimensional energy balance model.

## 3   Experimental stratgy

### 3.1   Experimental design

We use four AOGCMs (Table 1) with different qualities that are abruptly forced by 2X, 4X and 8X times the preindustrial $CO_2$

concentration. Some of the model experiments provide a longer integration time than 1000 years. However, we use only the first 1000 years of simulation time for consistency. The deep ocean adjusts on a multi-millennial timescale but the differences between the extrapolated and actual quasi-equilibrium temperature response at the Earth's surface are small. We analyze the global mean perturbations relative to the control state and use the changes in the surface air temperature $T(t)$ and net TOA imbalance $N(t)$ over time $t$ in order to generate understanding on the slow mode's behavior. The time series are based on

annual means.

During the course of the study we use the two-region framework (Eq. 4, 5) to interpret the temperature and radiative response of the slow mode. That is, $T = (1 - \chi)T_F + \chi T_S$ and $N = (1 - \chi)N_F + \chi N_S$. Having explored the separation of the fast and slow mode in the AOGCMs, we separate them consistently at year 21. At $t = 0$, the radiative forcing $F$ is equal to

$N_S(t = 0)$ without applying $\chi$, which is then equivalent to $N(t = 0)$. According to our conceptual framework Eq. (4,5), we assume that the fast mode and the slow mode are forced by the same global radiative forcing $F$. Thus, we compute the effective area weighting $\chi$ by the ratio between the global mean radiative forcing $F$ and the effective forcing of the slow mode $F_S$ which is the y-intercept in the statespace of $\chi T_S$ and $\chi N_S$.

A preliminary analysis reveals that model 1 and model 2 have positive temperature dependence of the global feedback, model 3 has zero or slightly negative feedback temperature dependence, and model 4 has negative feedback temperature dependence. We solved Eq. (1) for the three different forcing levels; that is, we use the 2X, 4X and 8X AOGCM experiments ($i$) and solve for $F_i = (\lambda + aT(\infty)_i)T(\infty)_i$. Having six equations, we rearrange first for the background feeedback parameter $\lambda$ and then, having three equations for each forcing level, for the coefficient for feedback temperature dependence $a$. Later on, we

rearrange for the background feedback parameter of the slow mode $\lambda_S$ and the associated coefficient for feedback temperature





**Table 1.** the AOGCMs considered in this study

| Model | Reference |
|---|---|
| Model 1 (positive feedback temperature dependence*) | MPIESM12 (Mauritsen et al., 2018; Rohrschneider et al., 2019) |
| Model 2 (positive feedback temperature dependence*) | HadCM3L (Cox et al., 2000; Cao et al., 2016) |
| Model 3 (zero feedback temperature dependence*) | CCSM3 (Yaeger et al., 2006; Danabasoglu and Gent, 2009) |
| Model 4 (negative feedback temperature dependence*) | CESM104 (Gent et al., 2011; Rugenstein et al., 2016a) |

\* In a preliminary analysis we solve Eq. (1) to estimate the feedback temperature dependence of the global feedback.

The radiative forcing $F$ is estimated by linearly regressing the relationship between $T$ and $N$ using years 1-10,

and the equilibrium temperature response $T(\infty)$ is estimated by linearly extrpolating this relationship using years 100-1000.

dependence $a_S$ (Eq. 5). Since our models exhibit a range of both negative and positive temperature dependence, we expect that we can generalize our analysis.

Using the different forcing levels, we use least-square fits of the zero-dimensional energy balance model (Eq. 5) to AOGCM
output in order to make predictions and address the deviations of these predictions from the AOGCM response. We expect that the AOCGMs differ in the magnitude and temporal adjustment of the slow mode $T_S$. The characteristic timescale of the slow mode should depend on the climate state while changes in ocean warming may modify the influence of feedback temperature dependence. The ocean response may result in considerable model spread.

### 3.2 Uncertainties

The present study provides arguments which are based on experimental results and does not assess long-term climate change in terms of quantifying forcing and temperature perturbations with great precision. However, during the course of the study we use uncertainties in the radiative forcing $F$ ($p^{50}$, $p^5$ $p^{95}$) and the slow mode's equilibrium temperature response $T_S(\infty)$ ($p^{50}$, $p^5$ $p^{95}$) in terms of percentiles in order to support our conceptual inferences. We compute the radiative forcing $F$ in the AOGCM experiments by linearly regressing $N$ against $T$. Using the first year as the lower end, we vary the upper end of the
regression time series (after yr 5 to year 20) and apply subsequently bootstrapping by replacement of the forcing estimates in order to generate the details of a continuous probability distribution. There is no unique way to determine the uncertainties in the radiative forcing $F$, because the estimate of $F$ is based on a sequence with respect to the evolution of $T$ and $N$. In this connection, the uncertainties themselves are subjective and an indication only. An alternative approach is the application of





bootstrapping by replacement before regressing the relationship between $N$ and $T$. Both approaches are biased, which points

out the lack of a sophisticated procedure to determine forcing uncertainties in a more precise way. The disadvantage of our

approach is that it overweights the first years of the regression time series as they are steadily involved in the regression. In this

sense, we likely overestimate the uncertainty, since the first years of the time series $T(t)$ and $N(t)$ are characterized by rapid

adjustments and they are strongly influenced by internal variability. Considering the upper limit of the regression length, our

approach neglects these first years because of that influence of major adjustments and internal variability of the climate system

at an initial stage. We choose year 20 as the maximum regression length because this timescale approximately describes the

equilibration of the fast mode in abrupt $CO_2$ experiments with AOGCMs.

According to the two-region framework (Eq. 4,5), we estimate the effective area weighting $\chi$ in order to scale the responses

of the slow mode in relation to the global mean forcing input $F$. We compute the effective area weighting $\chi$ by the ratio

between the global mean radiative forcing $F$ and the effective forcing of the slow mode $F_S$. We estimate the latter by linearly

regressing the relationship between the slow mode's temperature $\chi T_S$ and energy budget $\chi N_S$, using the years 21-120 as

regression time series to avoid the influence of feedback temperature dependence in the long-term. In general, the signal-to-

noise ratio is enhanced after the equilibration of the fast mode and we neglect uncertainties in $F_S$. That is, uncertainties in $F$

are translated into uncertainties in $\chi$ only, and in this way the estimates of the background feedback parameter $\lambda_S$ and feedback

temperature dependence $a_S$ are robust. As with the zero-dimensional energy balance model (Eq. 5), we interpret $\chi$ as effective

area weighting, although it is not possible to directly prescribe a spatial distribution due to the complex nature of the system's

response. Finally, we compute the uncertainties in $T_S(\infty)$ by linearly extrapolating the relationship between the slow mode's

temperature response $T_S$ and net TOA imbalance $N_S$. Using 10-year intervals, we increase the lower end (from year 100 to year

600) of the regression time-series and use year 1000 as the upper end. Subsequently, we apply bootstrapping by replacement

of the temperature estimates only to make sure that the distribution is not biased with respect to extreme outliers. In terms of

probabilities, the differences between the original sample and the posterior distribution are marginal.

### 3.3    AOGCM properties

In general, recent AOGCMs agree in that the Southern Ocean and the Eastern Tropical Pacific contribute substantially to

the emergence of the slow mode (not shown). These regions are directly coupled to the state of the intermediate and deep

ocean. However, nonlinear behavior of the slow mode is likely attributed to both local and nonlocal feedbacks as well as

state-dependent changes in the ocean component of the Earth system. In some models, nonlinearities may be highly localized,

whereas in other models nonlinear changes with forcing may be evenly distributed over the Earth's surface. We therefore ana-

lyze the slow mode's response from a global perspective.

The slow mode emerges from ocean heat uptake but also actuates a different feedback parameter than the fast mode in many

climate models, since the fast mode and the slow mode are not only discrete in terms of their temperature perturbation but

also in terms of their relationship to the net TOA radiative imbalance. This becomes evident when analyzing the relationship



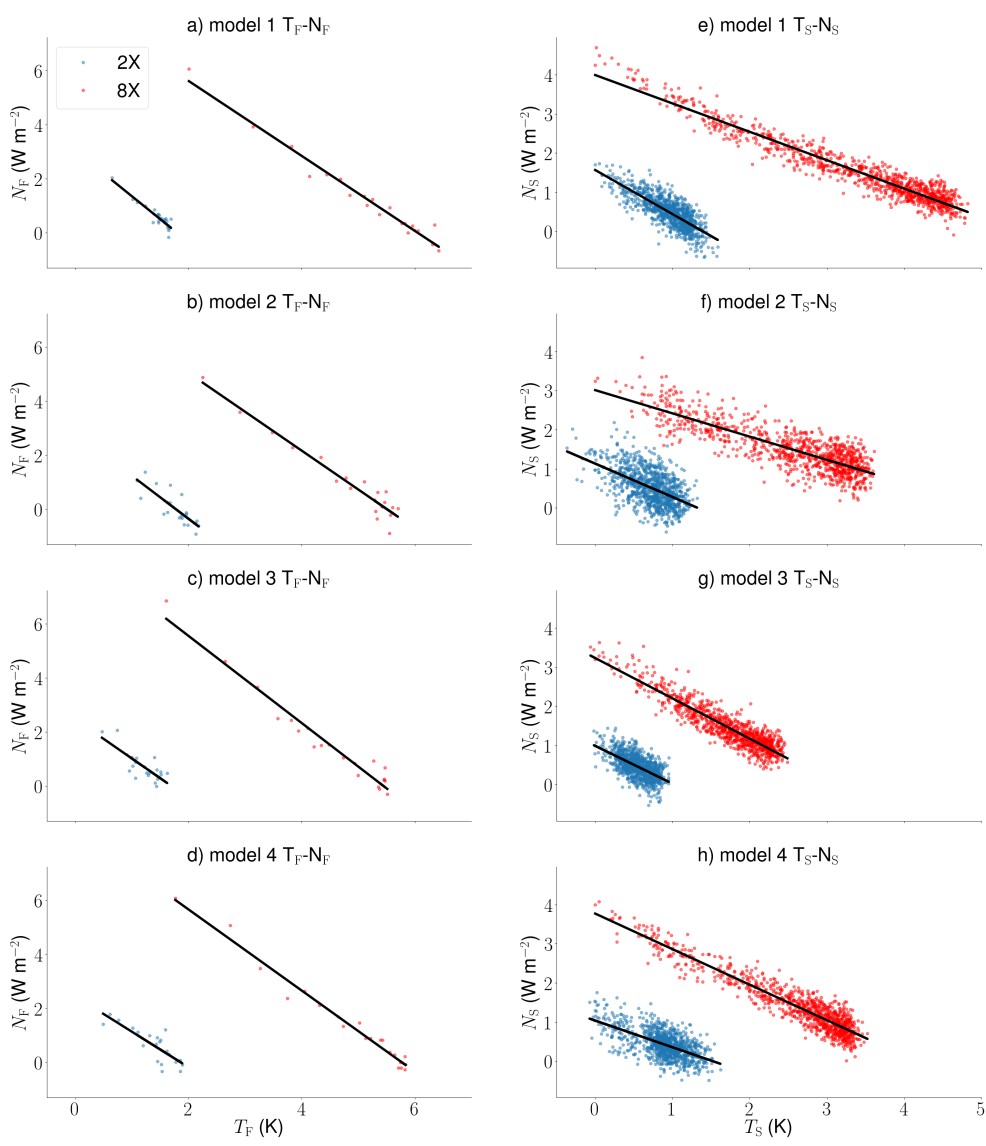

**Figure 2.** On the left (a,b,c,d), the relationship between the temperature perturbation $(1-\chi)T_F$ and net TOA imbalance $(1-\chi)N_F$ of the fast mode in the 2X (blue) and 8X (red) experiments. On the right (e,f,g,h), the relationship between the temperature perturbation $\chi T_S$ and net TOA imbalance $\chi N_S$ of the slow mode in the 2X (blue) and 8X (red) experiments. We further show the linear regression of the relationship between these variables in order to indicate the fast mode's feedback parameters $\frac{dN_F}{dT_F}$ and the slow mode's feedback parameter $\frac{dN_S}{dT_S}$ over the years 21-1000.





between $T$ and $N$ found in the AOGCM experiments (Fig. 2). As stated above, we interpret the relationship between between $T$ and $N$ as $T = (1-\chi)T_F + \chi T_S$ and $N = (1-\chi)N_F + \chi N_S$. In general, the relationship between $T_F$ and $N_F$ differs substan-

tially from the relationship between $T_S$ and $N_S$. In Fig. 2, we illustrate the magnitude of the fast mode's feedback parameter $\frac{dN_F}{dT_F}$ and the slow mode's feedback parameter $\frac{dN_S}{dT_S}$ by linear regression. The feedback parameter of the fast mode $T_F$ is much stronger than the feedback parameter of the slow mode $T_S$. Focusing on the slow mode, we find considerable changes in $\frac{dN_S}{dT_S}$ with forcing, and these changes are in line with the sign of the temperature dependence of the global feedback (Table 1). Comparing the 2X and 8X experiments, we find a decrease in the magnitude of $\frac{dN_S}{dT_S}$ with forcing in model 1 and model 2.

The feedback parameter $\frac{dN_S}{dT_S}$ stays approximately constant or increases slightly with forcing in model 3, and it increases with forcing in model 4. The nature of the relationship between $N$ and $T$ and the separation of the fast and slow mode makes it possible to apply the zero-dimensional energy balance model (Eq. 5) and analyze the temporal temperature adjustment of the slow mode in terms of parameters.

## 4   The influence of feedback temperature dependence

### 4.1   The slow mode in AOGCM experiments

We now analyze the slow mode's behavior by analyzing the temporal evolution of the global mean surface air temperature $T$. Fig. 3 (a,b,c,d) shows the evolution of the slow mode $T_S$ at the Earth's surface in the different AOGCMs. In model 1 and model 2, the slow mode increases nonlinearly with each doubling of the atmospheric $CO_2$ concentration, in the sense that

the relative temperature change between the $CO_2$ levels increases. By contrast, the relative temperature change between the different $CO_2$ levels stays approximately constant in model 3 and decreases in model 4. These changes correspond to the temperature dependence of the global feedback in the AOGCMs (Table 1). Next, we normalize the temperature adjustment of the slow mode $T_S$ by the radiative forcing $F$. Fig. 3 (e,f,g,h) shows the normalized evolution of the slow mode at the Earth's surface to demonstrate that the nonlinear behavior is mostly related to feedback temperature dependence. We use the the 2X

and 8X abrupt $CO_2$ experiment only in order to have a large signal. The nonlinear behavior of the slow mode $T_S$ in model 1 and model 2 does not arise from the changes in the radiative forcing $F$. The slow mode in model 3 behaves in a linear way and the lines for the temperature evolution are approximately congruent. Model 4 has negative feedback temperature dependence, and the normalized temperature response $T_S$ in the 2X experiment is higher than the normalized temperature response in the 8X experiment. In the following, we analyze the changes in the steady and temporal behavior of the slow mode in more detail.

Therefore, we characterize both the equilibrium response and timescale of the slow mode $T_S$ as represented in the AOGCMs.



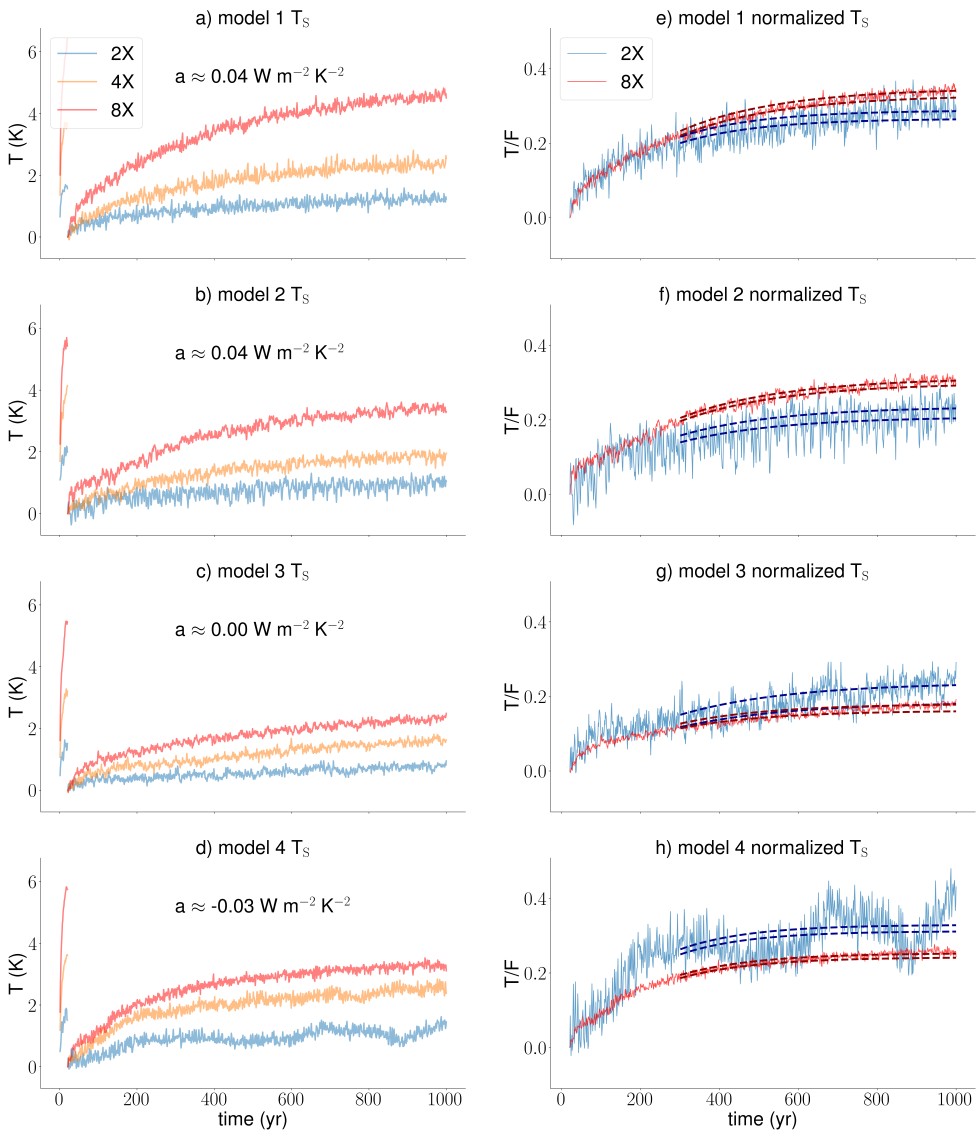

**Figure 3.** The slow mode $\chi T_{\mathrm{S}}(t)$ of the climate response in the AOGCMs experiments. On the left (a,b,c,d), we show the slow mode in the 2X (blue), 4X (orange) and 8X (red) abrupt $CO_2$ experiments. We also indicate the fast mode. On the right (e,f,g,h), we normalize the slow mode of the 2X (blue) and 8X (red) experiments by the forcing $F$ ($p^{50}$), and we further show the uncertainties ($p^5$,$p^{95}$) (darkblue and darkred) in the normalization.



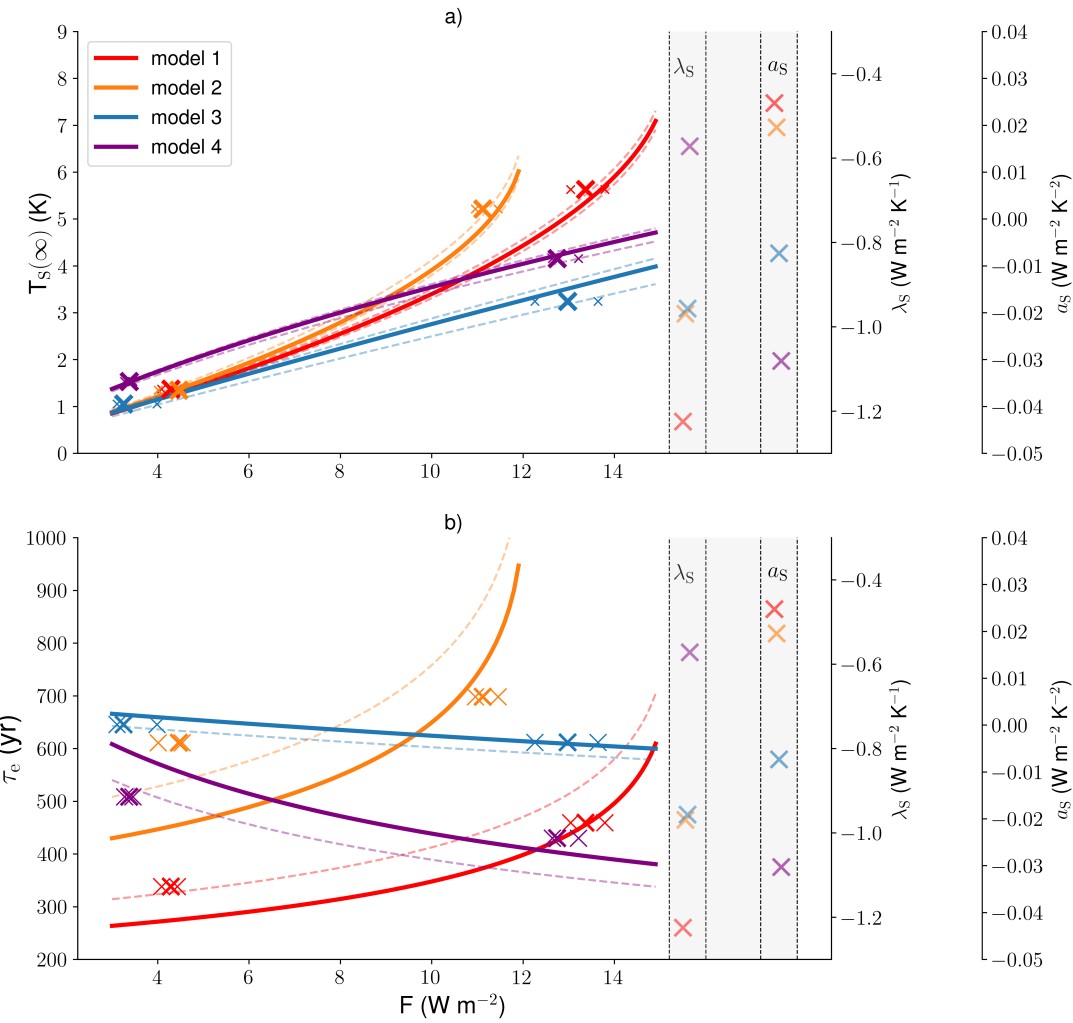

**Figure 4.** a) The equilibrium temperature response $\chi T_S(\infty)$ ($p^{50}$), and b) the approximated single mode e-folding timescale $\tau_e$ (Eq. 10) associated with the slow mode as a function of forcing. The x-markers represent the AOGCM output (2X,8X) and are based on the estimated forcing input $F$ ($p^{50}, p^5, p^{95}$). The lines are the predictions by the zero-dimensional energy balance model (Eq. 5) using $F(p^{50})$ and $T_S(\infty)(p^{50})$. In (a), we use the effective area weighting $\chi$ associated with the global mean forcing $F$ ($p^{50}$, solid) and $F$ ($p^5$ and $p^{95}$, dashed), computing the average over the three experiments. In (b), we fit the temperature evolution of the zero-dimensional energy balance model (Eq. 5) to the AOGCM experiments (2X,8X, solid) and (2X,4X,8X, dashed) in order to determine the effective heat capacity $C_S$. The background feedback parameters $\lambda_S$ and the feedback temperature dependencies $a_S$ are shown on the right.



## 4.2 The equilibrium response and timescale

We use the 2X, 4X and 8X AOGCM experiments ($i$) and solve for $F_i = (\lambda_S + a_S T_S(\infty)_i) T_S(\infty)_i$ in order to first compute the slow mode's background feedback parameter $\lambda_S$ and then the coefficient for feedback temperature dependence $a_S$. In this

way, the response of the slow mode $T_S$ can be described as a continuous function of the global mean radiative forcing $F$ (Fig. 4). The uncertainties in $F$ are translated into uncertainties in $\chi$ while the uncertainties in $\lambda_S$ and $a_S$ are marginal. In general, the background feedback parameter of the slow mode is less strong than the feedback of the fast mode and the net global feedback, which can be inferred from the relationship between $N$ and $T$ (Fig. 2). That is, the feedback temperature dependence $a_S$ associated with the slow mode is also less strong than the feedback temperature dependence of the global feedback. Fig. 4a

shows the equilibrium response of the slow mode $T_S(\infty)$ as a function of the radiative forcing $F$. We find that the equilibrium response $T_S(\infty)$ in the AOGCMs is well described by the zero-dimensional model with feedback temperature dependence. The equilibrium response $T_S(\infty)$ changes exponentially with an increase in forcing in the case of positive feedback temperature dependence. In the case of negative feedback temperature dependence, the response $T_S(\infty)$ is saturated with increasing $F$.

As with the simple models, we describe the changes in the temporal adjustment of the slow mode with forcing by the changes in the fitted single mode e-folding timescale (Eq. 10) (Fig. 4b). To do so, we determine a single heat capacity by fitting the temperature evolution of the zero-dimensional energy balance model (Eq. 5) to the temperature evolution of the slow mode in the 2X and 8X AOGCM experiments (year 21-1000). That is, we determine the heat capacity $C_S$. In general, the sign of the changes in the timescale $\tau_e$ with forcing in the AOGCMs coincide with the predicted changes by the simple model. The

timescales are forcing-dependent because the feedback of the slow mode depends on temperature, and the timescales change exponentially with higher forcing in the case of positive feedback temperature dependence. The timescales $\tau_e$ slightly decrease with higher forcing in the case of negative feedback temperature dependence. Focusing on model 1 and model 2, the changes in the timescales $\tau_e$ with higher forcing $F$ are important and of O(100) with respect to the low-end (2X) and high-end (8X) forcing range. In general, it is more difficult to predict the timescales of the slow mode than the equilibrium response $T_S(\infty)$. In

this connection, including an additional experiment such as a quadrupling of the atmospheric $CO_2$ concentration (4X) changes the absolute values significantly (dashed line). This shows that the predictions of the effective timescale are not robust, and the temperature evolution of the slow mode may not exactly follow the relationship described by Eq. (9) considering that the ocean's response influences the temporal evolution of the slow mode. To a large extent, these changes explain the major deviations from the theoretical predictions by the zero-dimensional energy balance model (Eq. 5), and we can interpret them

as changes in the effective heat capacity $C_S$. In the next section, we explore to which degree the temporal adjustment of the slow mode $T_S$ depends on the strength of the forcing.

## 4.3 Imprint of feedback temperature dependence

We quantify the differences in the timing of the long-term warming in the AOGCMs between the low-end (2X) and the high-end forcing range (8X). We prescribe the slow mode's temporal adjustment of the 2X experiments to the 8X experiments by





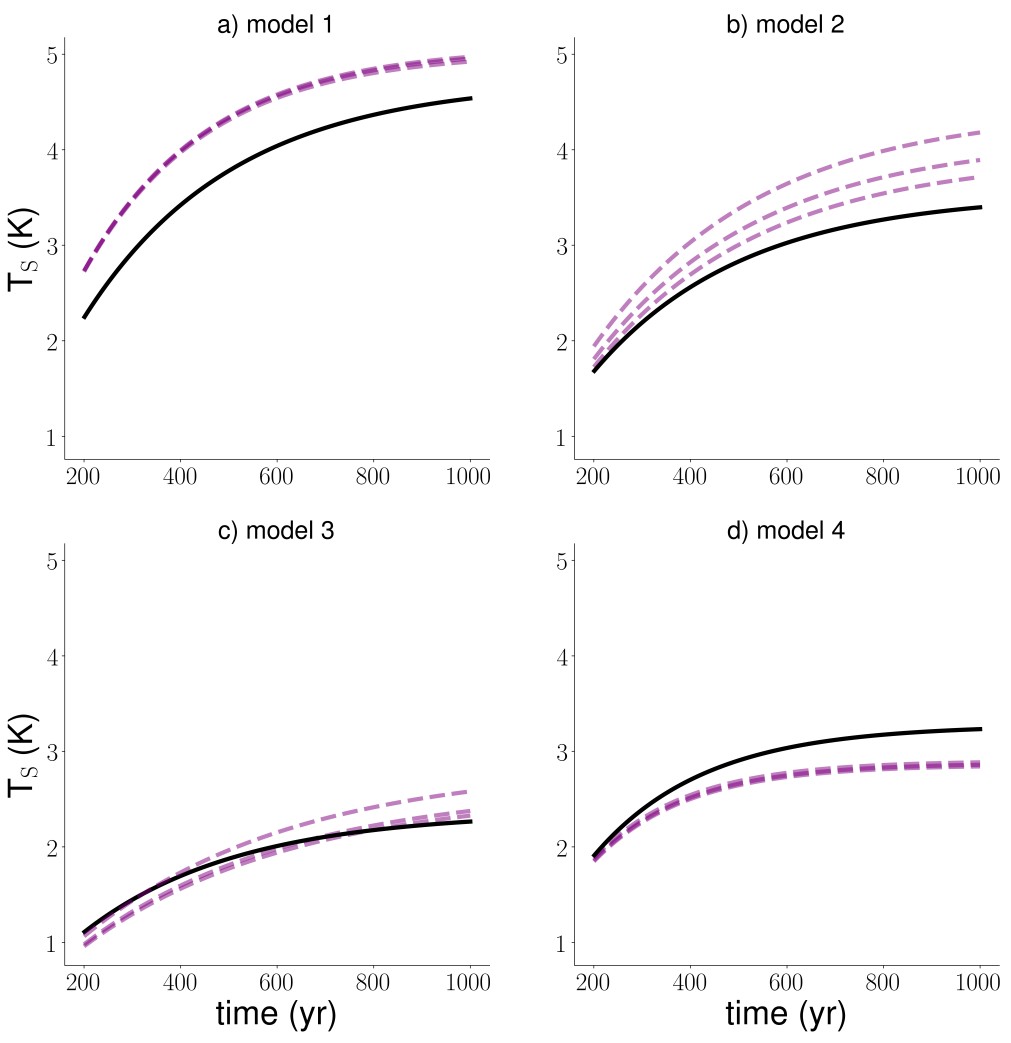

**Figure 5.** The slow mode $\chi T_{\mathrm{S}}(t)$ in the 8X AOGCM experiments (black) and the inferred slow mode $\chi T_{\mathrm{S}}(t)$ of the 8X experiments (purple) from the effective timescale $\tau(t)$ of the 2X experiment. We use the uncertainties that arise from the uncertainties $T_{\mathrm{S}}(\infty)$ ($p^{50}$, $p^5$, $p^{95}$). Using Eq. (11), we prescribe the temporal adjustment of the 2X experiment while using the equilibrium temperature response $T_{\mathrm{S}}(\infty)$ in the 8X simulations. In order to avoid internal variability, we fit an arbitrary e-folding mode, neglecting the initial adjustment and focussing on the temperature response on a centennial timescale.





using the equilibrium response $T_S(\infty)$ of the 8X experiments and the effective timescale of the 2X experiments. Assuming a linear e-folding mode, we quantify the effective timescale by rearranging Eq. (10);

$$\tau(t) = -t/log(\frac{T_S(\infty) - T_S(t)}{T_S.(\infty)}). \tag{11}$$

Fig. 5 shows that the temporal adjustment of the slow mode $T_S(t)$ between the 2X and 8X experiment is substantially altered. The computation is influenced by temporal changes of the effective timescale $\tau(t)$ in the 2X experiment such as changes in the

ocean circulation. However, we use uncertainties in $T_S(\infty)$ in order to support our inferences. The marked uncertainties support the perspective that the timescales change considerably due to the presence of feedback temperature dependence. Prescribing the temporal adjustment of the 2X experiment, we overestimate the temperature evolution on a multi-centennial timescale with higher forcing in the case of positive feedback temperature dependence (model 1 and model 2). We find only small changes in the case of approximately zero feedback temperature dependence (model 3), and we underestimate the temperature

evolution on a multi-centennial timescale with higher forcing in the case of negative feedback temperature dependence (model 4). In general, the influence of negative feedback temperature dependence is less strong than the influence of positive feedback temperature dependence. Our findings on the temporal adjustment of the slow mode in the AOGCMs indicate the importance of feedback temperature dependence for committed warming on a multi-centennial to millennial timescale in case of high forcing input. Linear models cannot capture long-term climate change in an appropriate way in the presence of feedback temperature

dependence, since they suggest that the effective timescale does not depend on the climate state, and the temporal adjustment of the slow mode would be independent of the degree of warming as found in model 3.

## 5   Varying timescale(s)

In this section we analyze the presence of both temperature or forcing-dependent and time-varying adjustment timescales $\tau(t)$. We further highlight the implications of the time-variation variation of $\tau(t)$ for eigenmode decomposition.

### 5.1   The effective time scale

We compare the effective timescale $\tau(t)$ (Eq. 11) of the slow mode found in the AOGCM experiments and the effective timescale $\tau(t)$ predicted by the zero-dimensional energy balance model (Eq. 5) (Fig. 6). The effective timescale describes the temporal temperature adjustment at any point of time. By its nature, $\tau(t)$ depends sensitively on the equilibrium response $T_S(\infty)$ in the sense that Eq. (11) is not an independent measure of the temporal behavior. However, it describes how tempera-

ture unfolds and therefore the fractions of the equilibrium response that are reached at different times. We likely underestimate $T_S(\infty)$ in the case of positive feedback temperature dependence and overestimate $T_S(\infty)$ in the case of negative feedback temperature dependence.

    As can be seen from Fig. 6, for both AOGCMs and the zero-dimensional energy balance model, the effective timescale $\tau(t)$

of the AOGCM experiments differs between the forcing levels, and these differences are in line with the feedback temperature





dependence of the slow mode $T_S$. The effective timescale $\tau(t)$ increases with higher forcing in the case of positive feedback temperature dependence (model 1 and model 2), stays approximately constant or decreases slightly in the case of model 3, and decreases in the case of negative feedback temperature dependence (model 4). The predicted timescale $\tau(t)$ of the zero-dimensional energy balance model (Eq. 5) shows an increase with time in the case of positive feedback temperature dependence

and high forcing input. In mathematical terms, this increase in $\tau(t)$ with time is attributed to the more complicated analytical solution of the energy balance model than Eq. (10) as discussed in section 2. In the case of negative feedback temperature dependence, the effective timescale $\tau(t)$ decreases slightly with time in the simple model.

By contrast, the AOGCM experiments reveal that the effective timescale $\tau(t)$ increases with time regardless of positive

or negative feedback temperature dependence or low and high forcing input $F$. The time-variation of $\tau(t)$ of the slow mode in the complex climate models is stronger than the time-variation predicted by the zero-dimensional energy balance model (Eq. 5). The uncertainties in the equilibrium response $T_S(\infty)$ support common model behavior, and even lower values than the estimated equilibrium response $T_S(\infty)$ would cause $\tau(t)$ to vary over time. In general, the temperature of the slow mode adjusts on longer timescales as time increases. The findings on $\tau(t)$ in the AOGCMs suggest a time-dependent component of the Earth

system that changes the inertia in the long-term such as changes in the ocean circulation. Thus, the temperature response of the slow mode does not exactly satisfy $C_S\frac{dT_S}{dt}$ (Eq. 5,9) in the sense that the effective heat capacity is no longer constant over time according to this framework, or suggesting a multiple mode structure. The time-varying adjustment timescale can be approximated by a multiple timescale structure of the slow temperature response, or vice versa, a multiple timescale structure of the slow temperature response is described by a time-varying timescale. The signal that arises from feedback temperature

dependence gives a state-dependent adjustment timescale and is robust, but the long-term temperature adjustment is related to multiple modes or a continuously-varying timescale. There is the possibility that the separation of the fast mode and slow mode at a specific year is inappropriate, but we expect that the errors are small and we could not explain the deviations from the simple model predictions.

## 5.2 Limits of the two-timescale approach

We illustrate the influence of the time-variation of $\tau(t)$ by focusing on the 8X experiments in order to explore the details of the slow mode's temperature adjustment in light of feedback temperature dependence $a_S$ and inconstant inertia $C_S$ (Fig. 7). We use a single forcing level only to highlight the limits of having a single exponential mode with a constant e-folding timescale and the zero-dimensional energy balance model also.

As a starting point, we assume a constant timescale $\tau$ which prescribes the temporal adjustment of a single e-folding mode with respect to the equilibrium response $T_S(\infty)$ (Eq. 10). The constant timescale $\tau$ represents the timescale $\tau(t)$ of the AOGCM experiments at an initial stage, using years 20-200. The temperature adjustment of the slow mode on multi-centennial timescale is considerably overestimated ($\Xi_1$), because the temperature adjustment of this exponential mode occurs on much shorter timescales than the actual timescales. Next, we compute analytically the exponential eigenmode by assuming that this eigen-





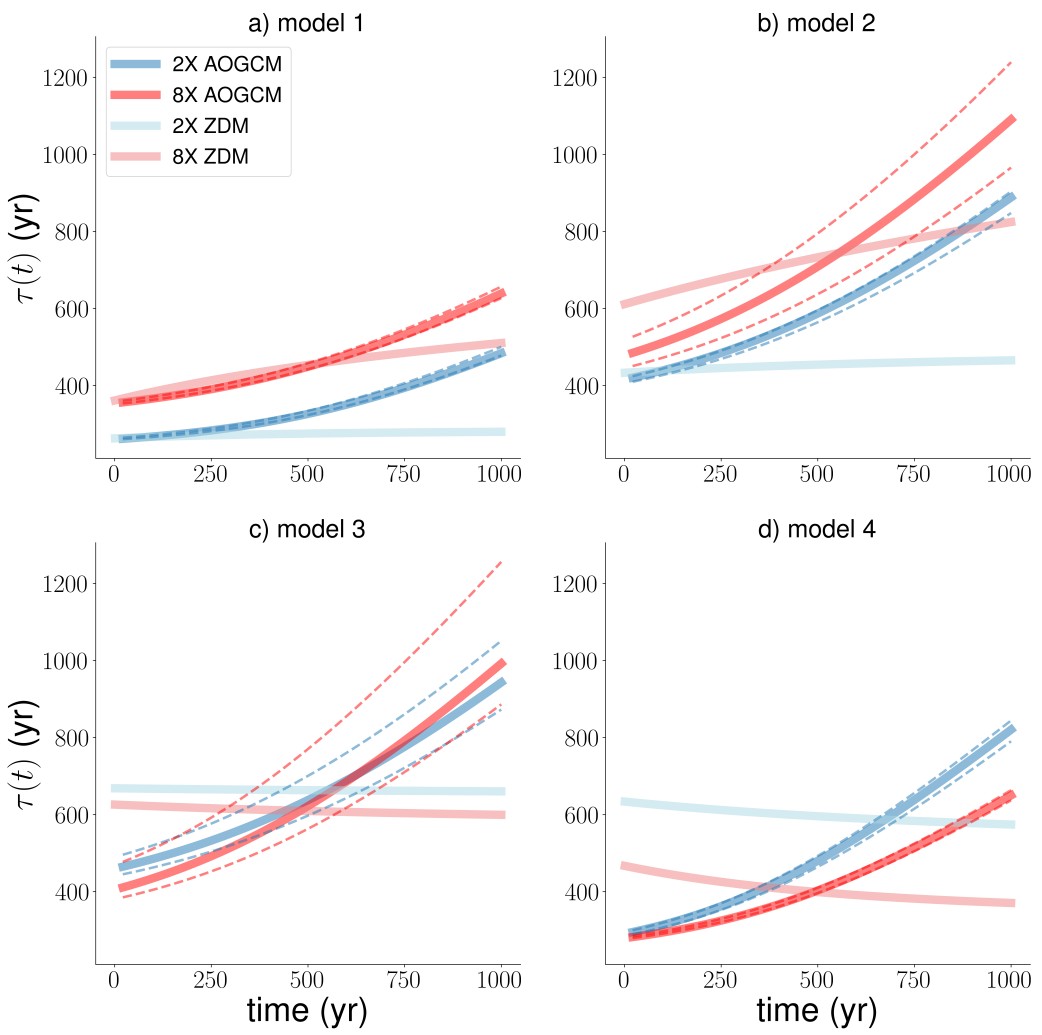

**Figure 6.** The effective timescale $\tau(t)$ (Eq. 11) of the slow mode $T_S(t)$ in the 2X (blue) and 8X (red) AOGCM experiments. We further show the predictions by a zero-dimensional energy balance model (ZDM) fit to the 2X experiments (orange) and 8X experiments (purple). Using the AOGCM output, we fit an arbitrary e-folding mode to exclude internal variability. Using $F(p^{50})$ and $T_S(\infty)(p^{50})$, the temperature evolution of the zero-dimensional energy balance model (Eq. 5) is fitted to the 2X and 8X AOGCM experiments in order to determine the effective heat capacity $C_S$. We recalculate the effective timescale $\tau(t)$ in the AOGCM experiments using the uncertainties that arise from the uncertainties $T_S(\infty)$ ($p^{50}, p^5, p^{95}$).

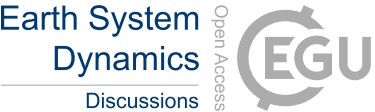

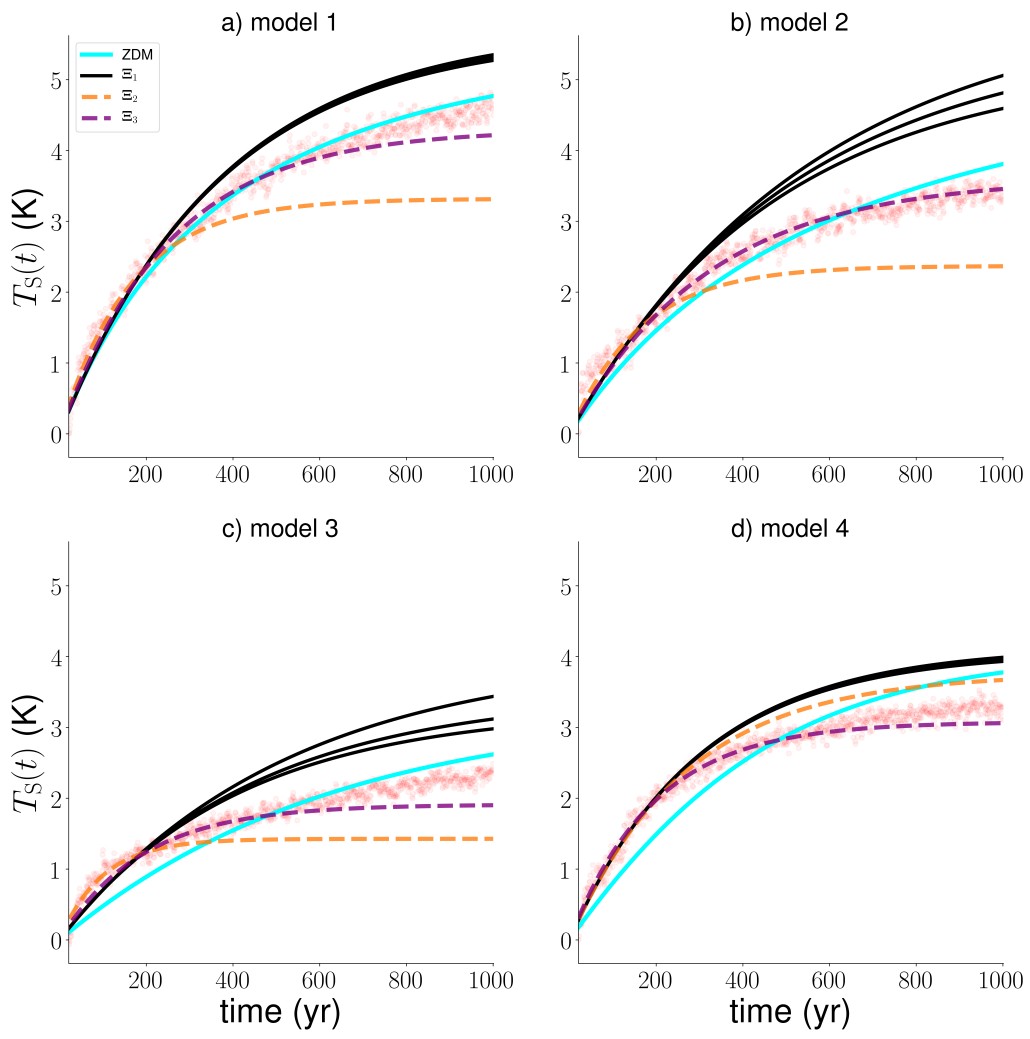

**Figure 7.** The temporal adjustment of the slow mode $\chi T_S(t)$ in terms of modes in the 8X AOGCM experiments (red). We show a single linear eigenmode (e-folding mode) with a constant timescale approaching the equilibrium response $T_S(\infty)$ ($p^{50}, p^5, p^{95}$), using the temporal average of years 20-200 of the effective timescale to compute $\tau$ at an initial stage ($\Xi_1$, black lines). We further show the analytical computation of the eigenmode (e-folding mode) using the time interval $t_1 = 100$ and $t_2 = 200$ ($\Xi_2$, dashed orange lines) and the time interval $t_1 = 200$ and $t_2 = 400$ ($\Xi_3$, dashed purple lines). For $\Xi_2$ and $\Xi_3$, the time series $\chi T_S(t)$ has been smoothed by a 100-year running mean. Finally, we show the temperature evolution of the zero-dimensional energy balance model (ZDM) (Eq. 5) fitted to the 8X AOGCM experiments ($F(p^{50})$ and $T_S(\infty)(p^{50})$) only in order to compute the effective heat capacity $C_S$ (cyan lines).



mode reproduces the temperature response in the time-interval $t_1$ and $t_2$. Using Eq. (10) we solve for a certain time-interval which gives the e-folding mode amplitude

$$T_{\Xi}(\infty) = \frac{T_{\mathrm{S}}(t_1)^2}{2T_{\mathrm{S}}(t_1) - T_{\mathrm{S}}(t_2)} \tag{12}$$

as long as $t_2/t_1 = 2$. We then solve Eq. (10) to compute the associated timescale $\tau_{\Xi}$. Considering the adjustment at an initial stage ($t_1 = 100$ and $t_2 = 200$), the temperature evolution of the eigenmode deviates strongly from the temperature evolution

in the 8X experiments in terms of amplitude and temporal adjustment ($\Xi_2$). The AOGCM temperature response and the eigenmode converge when using a time-interval on a multi-centennial timescale ($t_1 = 200$ and $t_2 = 400$), with a more accurate prediction with longer time-intervals ($\Xi_3$). Using longer time-intervals on a multi-centennial timescale is a circumvention to keep a single e-folding mode as a description for the slow mode's behavior. In doing so, errors are weighted to the long-term response while deviations are small on a multi-decadal and centennial timescale. For instance, choosing the time-interval

$t_1 = 500$ and $t_2 = 1000$ minimizes the errors considering the 1000-year temperature time series but the amplitude of these eigenmodes are substantially lower than the extrapolated equilibrium responses $T_{\mathrm{S}}(\infty)$ in the AOGCMs (not shown), using the relationship between $N$ and $T$.

Finally, the temperature evolution of the zero-dimensional energy balance model (Eq. 5) with feedback temperature depen-

dence is fitted to the temperature evolution in the 8X AOGCM experiments only. We underestimate the temperature response on a multi-decadal and centennial timescale and overestimate the temperature response on a multi-centennial timescale. The differences are small in the case of model 1 because the time-variation of $\tau(t)$ in model 1 is relatively low compared to model 2, model 3, and model 4. However, according to the zero-dimensional energy balance model (Eq. 5,9), $C_S$ increases over time, with much higher values on a multi-centennial timescale.

**6  Discussion**

Several studies use linear eigenmode decomposition and realize fitting (Eq. 3) in order to approximate the temperature evolution in response to radiative forcing (Olivie et al., 2012; Caldeira and Myhrvold, 2013; Proistosescu and Huybers, 2017). These studies are based on AOGCM simulations which have an integration time no longer than 300 years and thus do not capture properly the timescales of long-term climate change. In line with theory, Olivie et al. (2012) find that the temperature evolution

is described by a fast e-folding mode and slow e-folding mode. Caldeira and Myhrvold (2013) show that fitting two exponential modes with different timescales does equivalently reproduce the temperature response in AOGCMs as a one-dimensional slab diffusion ocean model does. In this connection, fitting two exponential modes does not necessarily imply two or more underlying and discrete processes, but it is ultimately based on a physical conceptual model. They further demonstrate that the difference between fitting two exponential modes and fitting three exponential modes is small on the timescale considered in

their analysis. From their perspective, fitting three exponential modes lacks an underlying physical theory. Finally, Proistosescu and Huybers (2017) find that three e-folding modes approximate the temperature evolution found in AOGCMs in more detail.



According to their analysis, a fast and intermediate exponential mode project primarily onto continental regions, whereas the slow mode on a centennial timescale is associated with the ocean adjustment. We consider the slow mode only and demonstrate that the timescale of the slow mode is forcing-dependent. We find that the temperature response at different forcing levels cannot

be described by an exponential eigenmode with a constant e-folding timescale. At the same time, however, a forcing-dependent timescale does not fully account for the time-variation of the effective timescale found in the AOGCMs considered here. Using linear eigenmode decomposition, at least one additional mode or a multiple mode and timescale structure of the slow adjustment is necessary to reproduce the details of long-term climate change in an appropriate way. Keeping a single mode, computing the eigenmode for the long-term response of the slow mode on a multi-centennial timescales circumvents a multiple mode and

timescale structure at the expense of a detailed representation of the temperature adjustment on a multi-decadal and centennial timescale and the actual equilibrium temperature response. Either way, such an eigenmode decomposition is valid for a certain forcing level only.

## 7 Summary and Conclusion

Feedback temperature dependence influences the slow mode of the climate response in a substantial way by changing both

the equilibrium response and timescale of long-term climate change which is thereby state-dependent. At the same time, the thermal inertia of the slow mode is not constant over time. Applying eigenmode decomposition, the two-timescale approach with constant timescales cannot capture the details of long-term climate change according to the experimental findings of the present study. The specific estimate of the timescale depends sensitively on the estimate of the equilibrium temperature response, but the finding that it changes with forcing and time is robust.


  The equilibrium response of the slow mode is well described by a zero-dimensional energy balance model that incorporates a background feedback parameter and a coefficient for feedback temperature dependence and a constant heat capacity. This model can be interpreted as an effective region which has much more inertia than the effective region that is associated with the fast mode. The zero-dimensional model captures major changes in the slow mode's adjustment timescale with forcing, and

the AOGCM experiments show that the temporal adjustment of the slow mode depends on the climate state in the case of non-zero feedback temperature dependence. However, the adjustment timescales predicted by the zero-dimensional model and the adjustment timescales found in the AOGCM experiments differ considerably. There is a stronger time-variation of the effective timescale of the slow mode in the AOGCM experiments than predicted by theory. The effective timescale of the slow mode in the AOGCM experiments increases with time regardless of positive or negative feedback temperature dependence. Thus, the

time-variation of the effective timescale cannot be explained by feedback temperature dependence only and the details of the slow, long-term temperature evolution are not well captured by the zero-dimensional energy balance model. Accordingly, the inconstancy of the thermal inertia can be approximated by a continuously varying timescale or a multiple timescale structure.



We find substantial model spread in how the AOGCMs reproduce long-term climate change, and state-dependent changes in the ocean's response may change the timing of the long-term temperature adjustment considerably. Our results depend on the outcome of a limited number of AOGCM experiments which are the only publicly experiments which have an integration time of 1000 years. The present study makes clear the importance of long-term climate change experiments simulated beyond year 2100 in order to predict and constraint the slow mode's behavior and future warming. However, both conceptual models and the AOGCMs considered here imply that feedback temperature dependence plays a large role in determining the extent and timing of long-term global warming. Research has to be done on why the adjustment timescale of long-term global warming is not constant over time.

*Author contributions.* TR lead the research, developed the coding scripts, and wrote the present manuscript. JBJ and MR reviewed and commented on the manuscript and its content.

*Data availability.* Data will be published via MPG.PuRe. We publish the data associated with the present study during the external review process. The codes associated with the present study are available upon reasonable request.

*Competing interests.* The authors declare that they have no conflict of interest.

*Acknowledgements.* This work was funded by the Max Planck Society (MPG) and the International Max Planck Research School on Earth System Modelling (IMPRS ESM). This project also received funding from the European Research Council (ERC) under the European Union's Horizon 2020 research and innovation programme (grant agreement No 786427, project 'Couplet'). We further thank Deutsches Klima Rechenzentrum (DKRZ) for providing the computational resources. Finally, we used the Python package scikit-learn (Pedregosa et al., 2011) to apply bootstrapping by replacement.





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
