# Peer review of "Does feedback temperature dependence influence the slow mode of the climate response?"

_Earth System Dynamics, 2021_

## Author Comment (AC1)

REVIEW 1

Dear reviewer, first of all we would like to thank you for your effort. We were happy to address the major and minor comments. We hope the paper is publishable according to your opinion.

We believe it does not make sense to list all improvements as we enhanced the contextual knowledge at various occasions. Instead, we created a file that gives the difference between the latest draft and the final manuscript which is submitted with this report. Please review this file to get an overview of the changes made.

We use the following fonts to mark the comment by the reviewer, the reply by the author, and the citation from the source code of the manuscript:

COMMENT
`REPLY`
`CITITATION`

COMMENT

Many studies have shown that the global surface temperature response to abrupt forcing is well approximated by two (or sometimes three) e-folding timescales, with usually a single timescale representing the "slow" warming beyond a couple decades at least out to 150 years or so. The authors pose an interesting question of whether this slow adjustment is well approximated by a single timescale when considering longer simulations out to 1000 years, and whether that timescale changes with forcing level given nonlinearities in climate feedbacks (lambda) and time dependence of the climate's effective heat capacity (c). It's obvious that a single timescale shouldn't work for the slow adjustment if you assume it's given by something like tau = -C/lambda and allow c and lambda to change over time or with temperature. The authors demonstrate this using AOGCMs forced by 2x, 4x, and 8x CO2. In particular, they suggest that climate feedback nonlinearities (lambda changing with temperature) changes the slow mode adjustment timescale, and that changes in effective heat capacity also play a role, all of which makes sense.

I) Unclear writing

A more general introduction would be helpful. You really throw the readers into the deep end on L20-24 with two redundant sentences that don't really say what a "slow mode" is (long term climate change is determined by the slow mode, and the slow mode describes the temporal adjustment on long timescales). A better introduction would be something along the lines of what you have written on L51-55. And then a clear statement could be made about the aim of the study, e.g., the two-mode approximation has been shown to work well for single forcing level and out to 150 years, but it's unclear whether this still holds for longer timescales and multiple forcing levels given feedback nonlinearities.

REPLY

We changed the introduction accordingly. We introduce the two-timescale approach first and mention our hypothesis. We then provide reason for our hypothesis. Finally, we illustrate the study content by a simple energy balance model. We did further improvements. line 15-125

CITITATION

\section{Introduction}
Studies of climate change have long found that the response of surface warming to radiative forcing occurs at multiple timescales. These studies typically differentiate between a fast timescale of response, that occurs within the first decade and is associated with the thermal inertia of the ocean's mixed layer and land, and a slow timescale of response, that occurs over centuries and is associated with the thermal inertia of the intermediate and deeper ocean \citep[e.g.][]{dickinson1998}. The latter is denoted as slow mode. Many studies have sought to understand how this slow mode of warming will unfold in time. This understanding is critical to predicting long-term warming. Commonly the slow mode has been modeled as exponential decay to equilibrium, and the decay has been assumed to be constant relative to the forcing level. In this paper we analyze the slow mode of the climate response in light of feedback temperature dependence and inconstant global heat capacity using abrupt CO$_2$ experiments with Atmosphere-Ocean General Circulation models (AOGCMs). More precisely, we analyze the timing of long-term climate change and therefore the adjustment timescale of the slow mode, which varies with temperature or forcing and time according to our study. We define the variation of the adjustment timescale with temperature or forcing as being state-dependent, and we define the variation of the adjustment timescale with the evolution of time as being time-dependent. We use this phrasing throughout the study. That is, we put forward the idea that the adjustment timescale of long-term climate change on a centennial timescale is state- and time-dependent, with the surface air temperature being the state-variable.
\newline

In this study feedback temperature dependence describes how the radiative feedback of the climate system depends on the global mean surface temperature change. The global feedback is given by the derivative of the global mean energy budget $N$ with respect to the global mean surface air temperature pertubation $T$, $\frac{d N}{d T}$. Commonly, the global feedback is assumed to be constant. However, it may change with temperature and time. We find that the dependence of the radiative feedback on temperature makes the timing of long-term climate change state-dependent. We also find a systematic time-dependent component of the adjustment timescale of the slow mode. Time-dependence of the adjustment timescale is possible due to changes in ocean heat uptake efficiency and horizontal heat transport, or an inconstant global ocean heat capacity. Oceanic timescales or heat capacities are commonly assumed to be constant. However, the circulation of the deep ocean may change \citep[e.g.][]{knutti2015} and diffusive ocean heat uptake progresses over time \citep{hansen1985,wigley1985}, which can make the deep ocean effective heat capacity inconstant. Another approach for a time-dependent timescale from a global

perspective is that there exists a geographic timescale pattern that emerges from the ocean circulation or a spatial pattern of local heat capacities.
\newline

To illustrate the study content, we make use of a simple energy balance model. The simple energy balance model for the global mean surface temperature response to forcing is given by
$C \frac{\mathrm{d}T}{\mathrm{d}t}= F + \lambda T$
where $C$ is the effective heat capacity of the global system, $T$ is the surface temperature perturbation relative to a reference state, $F$ is the radiative forcing which is e.g. the forcing by atmospheric CO$_2$ or aerosols, and $\lambda$ is the feedback parameter. In equilibrium, it is called the linear forcing-feedback framework. It is a first-order differential equation which analytical solution is given by $T(t) = \frac{F}{-\lambda} (1-e^{-t/\tau})$. The equilibrium warming is $T(\infty)=\frac{F}{-\lambda}$ and thus linear in forcing. The temporal adjustment is described by the e-folding timescale $\tau=\frac{C}{-\lambda}$ at which the single e-folding mode unfolds. That is to say, the response times of the temperature adjustment depend on $C$ and $\lambda$. The stronger the feedback parameter or the smaller the effective heat capacity, the more rapidly the system adjusts to forcing. In fact, $\lambda$ itself may depend on temperature which gives rise to temperature or forcing-dependent adjustment timescales. At the same time, $C$ may not be constant but time-dependent due to the changes in the ocean circulation and diffusive ocean heat uptake in response to global warming, or a spatial pattern of local heat capacities that prescribes the thermal inertia geographically.
\newline

…

COMMENT

L55: By "associated with a radiative feedback" do you mean "different regions associated with different radiative feedbacks"

REPLY

We think that the slow mode actuates a radiative feedback that is associated with the slow processes of the climate response. The slow mode may be a function of the deep ocean component or the slow surface air temperature response and radiative feedbacks may be aggregated onto a region which represents the slow mode. We add the following in

order to fulfill your requirements: line 92-113

CITITATION

Recent studies suggest that the slow mode is either a function of the Earth's deep ocean component \citep[e.g.][]{held2010,winton2010,geoffroy2013,geoffroy2013b}, or associated with a radiative feedback \citep[e.g.][]{armour2013,pro2017}. Considering the former concept, heat is taken up by the ocean on a decadal timescale that supresses the surface air temperature increase initially. Subsequently, the slow mode evolves because the ocean warms. Considering the latter process, the slow processes of the climate response may be aggregated onto a region which represents the slow mode. Mathematically, these different concepts are equivalent \citep{rohrschneider2019}.

...

COMMENT

 L58-74: I was getting lost at times whether you were talking about a timescale tau or the effective heat capacity C, and what the relationship is between them. I suggest rewriting to give a simple example first: C dT/dt = F + lambda*T, in which case tau = -C/lambda. Then you can point out that tau would not be constant if C is time dependent, which of course it is in climate models, or if lambda changes (e.g., the pattern effect or nonlinearities such that the feedback becomes lambda + a*T).

REPLY

We were happy to introduce considerations on a simple energy balance model, which illustrate the study content. line 45-55

CITITATION

To illustrate the study content, we make use of a simple energy balance model. The simple energy balance model for the global mean surface temperature response to forcing is given by
$C \frac{\mathrm{d}T}{\mathrm{d}t}= F + \lambda T$
where $C$ is the effective heat capacity of the global system, $T$ is the surface temperature perturbation relative to a reference state, $F$ is the radiative forcing which is e.g. the forcing by atmospheric $CO_2$ or aerosols, and $\lambda$ is the feedback parameter. In equilibrium, it is called the linear forcing-feedback framework. It is a first-order differential equation which analytical solution is given by $T(t) = \frac{F}{-\lambda} (1-e^{-t/\tau})$. The equilibrium warming is $T(\infty)=\frac{F}{-\lambda}$ and thus linear in forcing. The temporal adjustment is described by the e-folding timescale $\tau=\frac{C}{-\lambda}$ at which the single e-folding mode unfolds. That is to say, the response times of the

temperature adjustment depend on $C$ and $\lambda$. The stronger the feedback
parameter or the smaller the effective heat capacity, the more rapidly the system
adjusts to forcing. In fact, $\lambda$ itself may depend on temperature which gives
rise to temperature or forcing-dependent adjustment timescales. At the same time,
$C$ may not be constant but time-dependent due to the changes in the ocean
circulation and diffusive ocean heat uptake in response to global warming, or a
spatial pattern of local heat capacities that prescribes the thermal inertia
geographically.
\newline

COMMENT

170 (Conceptual Insights): I found this section to be very confusing. You introduce a two-region model (equations 4 and 5) which later you use to analyze the slow response in GCMs. You could use this model to straightforwardly make your point that the response timescale (in this case tau_S = C_S/ (lambda_S+a_S*T_S)) is not constant if C_S changes or for nonzero a_S. But instead, you introduce the two-layer model which is mathematically equivalent to equations 4 and 5 for the case a_F=a_S=0 (as shown in Geoffroy et al. 2013) but appears to have a very different form. You then make the confusing statement that "However, the parameters of the two-layer model modify the inertia of the slow mode. For instance, parameter for the efficiency of ocean heat uptake eta is an inertia parameter, and changes in ocean heat uptake cause C_S to increase or decrease". It's unclear whether you mean that C_S can change with model parameters such as eta (which is obvious because C_S can be written as a function of eta), or whether you mean that C_S changes over time somehow (which is the topic of the paper, but not obvious from these equations). L156-167 confuse things further by introducing a new approximate definition of the slow mode which asserts a constant tau_S while noting that the full solution to equation (5) has a time varying C_s (presumably for the case of nonzero a?). But then Figure 1c suggests that tau_S is a constant function of a, which is confusing given that equation (5) suggests tau_S should change over time for nonzero a_S. //Overall, much more clarity is needed here about whether you are talking about C_S changing with model parameters or about changing over time. It's also not clear that discussing the 2-layer model adds anything at all given that it seems to confuse things and you don't use it outside of this section anyway.

REPLY

We add the following explanation in order to enhance the comprehensibility or contextual knowledge. We explain why we need the different models. We hope the extension meets your expectations. line 127-138 and 191-205

CITITATION

\section{Conceptual insights}
Before exploring the slow mode's behavior in AOGCMs, we provide conceptual insights about the slow mode using simple climate models. We explain conceptually the slow mode as well as demonstrate conceptually the state- and time-dependence of the adjustment timescale of the slow mode. The simple models considered here are energy balance models and outlined in detail in \citet{geoffroy2013}, \citet{geoffroy2013b}, \citet{armour2013}, \citet{rohrschneider2019}, among others. We bring together these existing concepts to lay out the parameter dependencies of the slow mode in order to provide a solid basis and motivation for our experimental analysis. With this section we provide insight how the fast e-folding mode and the slow e-folding mode (Eq. 3) emerge from simple assumptions using energy balance models. We present two recent concepts: the two-region framework which is used in this study to analyze the slow mode; and a two-layer model in which the slow mode is a function of the Earth's deep ocean component. The two-region model is much simpler than the two-layer model, while the two-layer model accounts explicitly for changes in the ocean circulation. However, the two-layer model can be expressed mathematically by the two-region model.
\newline

...

For the experimental analysis, we choose the two-region model because it provides a simple framework to understand the slow mode, allowing us to analyze the different parameters. By its nature, the two-layer model is more complicated, having more than one inertia parameter and an efficacy term. However, it is necessary to mention the two-layer model because the slow mode can also be understood as a function of the deep ocean component, and changes in ocean heat uptake or in the heat uptake efficiency do change the behavior of the slow mode. Heat uptake efficiency is an inertia parameter in the two-layer model and it changes the timing of climate change and the magnitude of the fast mode and the slow mode while the magnitude of the global equilibrium response remains unchanged. Heat transport efficiency is associated with the two-region model by changing the heat capacity of the region that influences the timing of climate change as well as the regional feedback parameter that gives the magnitude of the surface air temperature response. In the following we analyze briefly the parameter dependencies of the equilibrium response and timescale of the slow mode using the two-layer ocean model. We use the more complicated two-layer ocean model to show the dependence of the slow mode on the ocean circulation besides feedback temperature dependence. We focus on the heat uptake efficiency in the two-layer model to provide an explanation for changes in C$_S$. C$_S$, may change with model parameter, and this model parameter may also vary with time. It is not straightforward to find a simple analytical expression for the dependence of C$_S$ on $\eta$.
\newline

COMMENT

L237-239: I suggest you cite papers showing this

REPLY

We add the following: line 310-313

CITITATION

In general, recent AOGCMs agree in that the Southern Ocean and the Eastern Tropical
Pacific contribute substantially to the emergence of the slow mode. We did a
preliminary analysis of the outcome of the four AOGCMs used in this study to
confirm this (not shown).

COMMENT

L245: I don't follow this sentence. What do you mean by "the slow mode emerges from heat uptake"?

REPLY

Heat is taken up by the ocean on a decadal
timescale that surpresses the surface air
temperature increase initially. Subsequently, the
slow mode evolves because the ocean warms. We add
this to the text as well as much more contextual
knowledge.

COMMENT

L283-284: I don't understand this sentence. Doesn't a_S represent feedback temperature dependence on long timescales? Then how can it be less strong than that?

REPLY

We split the surface air temperature response and energy budget into a fast response and slow response such that the global response without splitting it into components differs from the fast response and the slow response. We add this to the text as well as much more contextual knowledge.

COMMENT

L302-305: Getting lost here. I think you are saying that your theoretical predictions don't work because the effective heat capacity is not constant, which makes sense. But can you show this, rather than simply suggesting it? Is there a way to account for changes in heat capacity separately from feedback nonlinearity in the response timescale?

REPLY

We implicitly do this later on when comparing the effective timescale of the different AOGCMs and the effective timescale of the simple model, section 5.1. Feedback temperature dependence introduces state-dependent changes in the timing of long-term climate change and cannot account for the full time-dependence of the adjustment timescale of the slow mode. The latter is rather associated with temporal changes in C$_S$. We add some more explanations.

COMMENT

L345-346: How are these "mathematical terms"? This is an unclear explanation that I don't follow

REPLY

Thank you. We mean the analytical solution of the differential equation. We add *formally, in relation to mathematical terms\** instead of only mathematical terms.

COMMENT

Discussion: I was hoping by this point to have some clarity about whether additional timescales are needed to model the slow response (beyond year 21). The results seem to suggest that a single timescale is not good enough, which makes sense. But is this true only because of feedback nonlinearities, or also because of effective heat capacity changes? Could theory be saved by adding one (or more) additional slow timescales? I do not know what the takeaways are here.

REPLY

We are going to state more explicitly the major takeaways throughout the manuscript. Assuming a slow mode, it depends on the climate state and the evolution of time. Both influence the timescale structure as explained below. We add the following: line 510-514

CITITATION

Summarizing, feedback temperature dependence introduces a state-dependent slow mode adjustment timescale which cannot be reproduced by a single timescale that is constant with respect to the forcing input. At the same time, the adjustment timescale is time-dependent due to inconstant $C_{\mathrm{S}}$ in the sense that a single constant timescale cannot reproduce the slow mode's temperature adjustment at a single forcing level. A time-dependent timescale can be approximated by a multiple timescale approach, whereas state-dependence implies that the timescale or the multiple timescale approach varies with forcing.

COMMENT

L434-436: This is an important summary point which greatly helps to clarify the purpose of your study. But I could not tell you where these points were shown in the paper. Which figures show this clearly?

REPLY

We believe now it is clear that the different figures provide this knowledge in an explicit way. Having various improvements on the contextual knowledge and a better introduction, we hope we made this point. The introduction provides a better background in the sense that the knowledge can be embedded.

COMMENT

L445-446: Wasn't this the point of this study?

REPLY

There are different reasons why there is time-dependence of the adjustment timescale of the slow mode. Feedback temperature dependence introduces a state-dependent response. We add the following line 542-546

CITITATION

However, both conceptual models and the AOGCMs considered here imply that feedback temperature dependence plays a large role in determining the extent and timing of long-term global warming. Research has to be done on why the adjustment timescale of long-term global warming is systematically time-dependent. It can be explained by both changes in the ocean circulation or a geographic pattern of local effective heat capacities that emerge from the ocean circulation.

II Unclear method

COMMENT

L181-189: A schematic would really help here. I think you are describing Gregory plots for the fast and slow mode, but I was not able to follow your methods without sketching out what the Gregory plots would look like for fast and slow modes separately and their combination via the

REPLY

We illustrate these lines by a schematic now. See Fig. (2). Please review the comment below.

COMMENT

L183-184: All of your results follow from this choice to separate fast and slow modes at year 21. How did you choose this separation year (other than following earlier papers, e.g., DOI: 10.1175/JCLI-D-14-00545.1)? What did you find when you "explored the separation of the fast and slow mode"? Do your results depend on this choice at all? And, how do you actually define T_S and N_S for the slow mode? Do you take values of T and N at year 21 and subtract them from all subsequent years of the T and N timeseries to calculate T_S and T_N, or something else?

REPLY

We add some more explanations and statements about the separation of the fast mode and the slow mode. A difference of one year in the separation does not make a significant difference  in the degree to which the regional radiative feedback depends on temperature. Below are the changes made which answer your questions. We provide a sketch also (Fig. 2). line 246-259

CITITATION

During the course of the study we use the two-region framework (Eq. 4, 5) to interpret the temperature and radiative response of the slow mode. That is, $T=(1-\chi) T_\mathrm{F}+\chi T_\mathrm{S}$ and $N=(1-\chi) N_\mathrm{F}+\chi N_\mathrm{S}$. Having explored the separation of the fast and slow mode in the AOGCMs, we separate them consistently at year 21. The fast mode in the first region $(1-\chi) T_\mathrm{F}$ is thus the temperature adjustment $T$ until year 21, whereas the slow model is given by the difference $\chi T_\mathrm{S}= T-(1-\chi) T_\mathrm{F}$. At $t=0$, the radiative forcing $F$ is equal to $N_\mathrm{S}(t=0)$ without applying $\chi$, which is then equivalent to $N(t=0)$. According to our conceptual framework Eq. (4,5), we assume that the fast mode and the slow mode are forced by the same global radiative forcing $F$. Thus, we compute the effective area weighting $\chi$ by the ratio between the global mean radiative forcing $F$ and the effective forcing of the slow mode $F_\mathrm{S}$ which is the y-intercept in the statespace of $\chi T_\mathrm{S}$ and $\chi N_\mathrm{S}$. Fig. 2 sketches the two-region framework considering the relationship between $N$ and $T$. Considering the separation at year 21, it is worth noting that a difference of one year in the separation does not make a significant difference in the AOGCM fitted feedback temperature dependencies and heat capacities of the two-region model. In this sense, our findings are robust and less influenced by the specific separation at year 21. The separation at year 21 is supported by the analysis of Andrews et al. (2015).
\newline

COMMENT

L190-195: More detail is needed here, for example: How do you define and calculate the "background feedback parameter lambda"? With what runs? Over what timescales? Is it assumed to not change between forcing levels, and is this a good assumption? I imagine that if different forcings produced different patterns of warming, that would result in different feedbacks from pattern effects rather than global temperature nonlinearities, but has this been accounted for somehow?

REPLY

We use the 2X, 4X, and 8X CO2 forcing levels to compute the feedback temperature dependence, using Eq. (1 and/or 5). We believe it is clearly written in the manuscript, with providing an equation. We add that it is computed by rearrangement of the parameters and how many equations we have.

COMMENT

How is T(infinity) estimated from transient simulations?

REPLY

By extrapolation of the surface air temperature with respect to zero energy imbalance. We add this to the text.

COMMENT

I got lost trying to figure out what the three equations were for each forcing level, and how you went about fitting for all the parameters. I suggest writing this out explicitly for readers to follow.

REPLY

We use Eq. (1) to obtain the feedback temperature dependence of the global feedback; and we use Eq. (5) under the two-region framework (Eq. 4,5) to obtain the feedback temperature dependence of the slow mode.

COMMENT

Do you assume that a_F and a_S are the same, or can they be different?

REPLY

They are different because we split the surface air temperature response and the energy budget into a fast mode and a slow mode. We add this explanation.

COMMENT

How do you calculate effective radiative forcing in all the runs? (you mention this later, but it should be stated clearly here.)

REPLY

By linear regression with respect to the y-intercept in the relationship between $T$ and $N$. We add this.

COMMENT

L208-211: I don't follow the method you describe. I understand you do regression over different lengths of years (from 5 to 20), but what does it mean to "apply subsequently bootstrapping by replacement of the forcing estimates in order to generate the details of a continuous probability distribution"?

REPLY

It is a resampling method, we add a definition of bootstrapping. We add the following: line 279-284

CITITATION

Using the first year as the lower end, we vary the upper end of the regression time series (after yr 5 to year 20) and apply subsequently bootstrapping by replacement of the forcing estimates in order to generate the details of a continuous probability distribution. Bootstrapping is a resampling technique by sampling a dataset with replacement. The original population is given by the regression estimates. There is no unique way to determine the uncertainties in the radiative forcing $F$, because the estimate of $F$ is based on a sequence with respect to the evolution of $T$ and $N$.

COMMENT

L222-229: Again, a schematic would help show what you mean here. It's not obvious how all this works without showing the reader.

Thank you. We hope the new explanations are
sufficient for general understanding now.
Otherwise, please let us know.

COMMENT

L233-236: Is it a linear extrapolation a good assumption given the feedback nonlinearities you find? How off might your estimates be? It's also unclear what you did with bootstrapping here again.

REPLY

It is conservative; the magnitude of feedback
temperature dependence is underestimated.

COMMENT

Figure 4: I did not get much out of Figure 4. What are we supposed to learn here? Are we to take away that the energy balance model predictions match the AOGCM responses or not?

REPLY

It is an important figure, we may add some more
explanations. Feedback temperature dependence is
easily calculated and a robust parameter, whereas
it is more difficult to fit the temporal behavior

found in AOGCMs. Important conclusions are drawn
from this finding such that C$_S$ is no robust
AOGCM model parameter. We add the following:
line 551-353

CITITATION

We fit Eq. 5 to AOGCM output. Feedback temperature dependence is easily computed
and a robust parameter, whereas it is more difficult to fit the temporal behavior
found in AOGCMs considering $C_{\mathrm{S}}$ when assuming that it is constant.
\newline

COMMENT

L278: What does it meant to solve for an equation? Do you mean a specific set of the parameters in the
equation? By what method?

REPLY

We have three forcing levels and two parameters.
Thus, we rearrange to get the parameters.

COMMENT

L341-343: It's plausible that it's differences in feedback temperature nonlinearity causing the
differences in the temporal changes in tau between models, as you suggest. But can you show this?
Could changes in effective heat capacity not also play a role?

REPLY

The comparison between the simple model and AOGCM
output in this section and the previous section

reveals that there are temporal changes that depend on forcing (climate state, feedback temperature dependence) and that there is time-dependence of the adjustment timescale that is independent of forcing. We add an explanation. line 409-425

CITITATION

\section{Varying timescale(s)}
In this section we analyze the presence of both the temperature or forcing-dependent and time-varying adjustment timescale $\tau (t)$. We further highlight the implications of the time-variation of $\tau(t)$ for eigenmode decomposition. We show that there is a time-dependent component in the adjustment timescale besides the state-dependence. The two-region framework with constant $C_{\mathrm{S}}$ can only reproduce the state-dependence of the temporal temperature adjustment.

\subsection{The effective time scale}
We compare the effective timescale $\tau (t)$ (Eq. 11) of the slow mode found in the AOGCM experiments and the effective timescale $\tau (t)$ predicted by the zero-dimensional energy balance model (Eq. 5) (Fig. 7), which demonstrates the state-dependent component and the time-dependent component of the adjustment timescale in a generic way. The effective timescale describes the temporal temperature adjustment at any point of time. By its nature, $\tau (t)$ depends sensitively on the equilibrium response $T_\mathrm{S} (\infty)$ in the sense that Eq. (11) is not an independent measure of the temporal behavior. However, it describes how temperature unfolds and therefore the fractions of the equilibrium response that are reached at different times. We likely underestimate $T_\mathrm{S} (\infty)$ in the case of positive feedback temperature dependence and overestimate $T_\mathrm{S} (\infty)$ in the case of negative feedback temperature dependence. With the previous section it is shown that the timing of long-term climate change depends on the climate state or forcing due to feedback temperature dependence. With this section we additionally point out that the adjustment timescales of the slow mode depend on the evolution of time. The difference between AOGCM output and simple model predictions highlights the importance of the time-dependent adjustment timescale to reproduce the details of the temperature evolution.
\newline

…

COMMENT

Figure 6: Again, it's unclear what to take away from this figure. That the energy balance model doesn't replicate the AOGCM output? What is learned?

REPLY

Let us say it describes the state-dependent component and the time-dependent component of the adjustment timescale in a generic way. We hope it is clear now with the explanations provided above and much more contextual knowledge.

COMMENT

Section 5.2: I have admittedly run out of steam here, but I don't know what the point of this section is or follow its methods.

REPLY

We analyze the degree to which only time-dependence matters with respect to the surface air temperature adjustment. We add explanations and a table.
Line 453-485

CITITATION

We illustrate the influence of the time-variation of $\tau (t)$ by focusing on the 8X experiments in order to explore the details of the slow mode's temperature adjustment in light of feedback temperature dependence $a_\mathrm{S}$ and inconstant inertia $C_\mathrm{S}$ (Fig. 8). We use a single forcing level only to highlight the limits of having a single exponential mode with a constant e-folding timescale and the zero-dimensional energy balance model also. We thus analyze the degree to which only the time-dependence of the adjustment timescale of the slow mode influences the surface air temperature response. In mathematical terms, we use different approaches and assumptions on the adjustment timescale to fit the temperature evolution of the slow mode. The different methods are summarized in Table 2 and provide an estimate of the importance of the time-variation of $\tau (t)$.
\newline

```latex
\begin{table}
% table caption is above the table
\caption{Methods to fit the temperature evolution of the of the 8X AOGCM
experiments }
\label{tab:1}        % Give a unique label
% For LaTeX tables use
\begin{tabular}{ll}
\noalign{\smallskip}\hline\noalign{\smallskip}
\textbf {fit} & \textbf {Description}\\
\\
$\Xi_\mathrm{1}$ & we use the effective timescale using years 20-200 to fit the
temperature evolution \\
\\
$\Xi_\mathrm{2}$ &  we combine Eq. (12) and Eq. (10) using the time interval
$t_1=100$ and $t_2=200$ to fit the temperature evolution    \\
\\
$\Xi_\mathrm{3}$&   we combine Eq. (12) and Eq. (10) using the time interval
$t_1=200$ and $t_2=400$ to fit the temperature evolution\\
\\
$ZDM$ & Eq. (5) is fitted to the temperature evolution\\
\\
\noalign{\smallskip}\hline
\end{tabular}
\end{table}
```

As a starting point, we assume a constant timescale $\tau$ which prescribes the
temporal adjustment of a single e-folding mode with respect to the equilibrium
response $T_\mathrm{S}(\infty)$ (Eq. 10). The constant timescale $\tau$ represents
the timescale $\tau(t)$ of the AOGCM experiments at an initial stage, using years
20-200. The temperature adjustment of the slow mode on multi-centennial timescale
is considerably overestimated ($\Xi_\mathrm{1}$), because the temperature
adjustment of this exponential mode occurs on much shorter timescales than the
actual timescales. Next, we compute analytically the exponential eigenmode by
assuming that this eigenmode reproduces the temperature response in the time-
interval $t_1$ and $t_2$.  Using Eq. (10) we solve for a certain time-interval
which gives the e-folding mode amplitude
\begin{equation}
T_\mathrm{\Xi}(\infty)=\frac{T_\mathrm{S}(t_1)^2}{2 T_\mathrm{S}(t_1) -T_\mathrm{S}(t_2)}
\end{equation} as long as $t_2/t_1=2$. We then solve Eq. (10) to compute the
associated timescale $\tau_\mathrm{\Xi}$. Considering the adjustment at an initial
stage ($t_1=100$ and $t_2=200$), the temperature evolution of the eigenmode
deviates strongly from the temperature evolution in the 8X experiments in terms of
amplitude and temporal adjustment ($\Xi_\mathrm{2}$). The AOGCM temperature
response and the eigenmode converge when using a time-interval on a multi-
centennial timescale ($t_1=200$ and $t_2=400$), with a more accurate prediction
with longer time-intervals ($\Xi_\mathrm{3}$). Using longer time-intervals on a
multi-centennial timescale is a circumvention to keep a single e-folding mode as a
description for the slow mode's behavior. In doing so, errors are weighted to the
long-term response while deviations are small on a multi-decadal and centennial
timescale. For instance, choosing the time-interval $t_1=500$ and $t_2=1000$
minimizes the errors considering the 1000-year temperature time series but the
amplitude of these eigenmodes are substantially lower than the extrapolated
equilibrium responses $T_\mathrm{S} (\infty)$ in the AOGCMs (not shown), using the
relationship between $N$ and $T$.
\newline

Finally, the temperature evolution of the zero-dimensional energy balance model
(Eq. 5) with feedback temperature dependence is fitted to the temperature evolution

in the 8X AOGCM experiments only ($ZDM$). We underestimate the temperature response on a multi-decadal and centennial timescale and overestimate the temperature response on a multi-centennial timescale. The differences are small in the case of model 1 because the time-variation of $\tau (t)$ in model 1 is relatively low compared to model 2, model 3, and model 4. However, according to the zero-dimensional energy balance model (Eq. 5,9), $C_{S}$ increases over time, with much higher values on a multi-centennial timescale.

---

## Author Comment (AC2)

REVIEW 2

Dear reviewer, first of all we would like to thank you for your effort. We were happy to address the major and minor comments. We hope the paper is publishable according to your opinion.

We believe it does not make sense to list all improvements as we enhanced the contextual knowledge at various occasions. Instead, we created a file that gives the difference between the latest draft and the final manuscript which is submitted with this report. Please review this file to get an overview of the changes made.

We use the following fonts to mark the comment by the reviewer, the reply by the author, and the citation from the source code of the manuscript:

`COMMENT`
`REPLY`
`CITITATION`

COMMENT
Summary:
This paper presents interesting and useful new results on the timescales of the climate response to CO2 forcing, exploiting 1000-year long step forcing AOGCM experiments. While the results are novel, I found the presentation rather complex and hard to follow, so I am requesting major revisions to make the paper more accessible.
Main comments:
Presentation:

Overall I found the text difficult to read, despite it being well polished and free of typos – to the point that I didn't understand everything despite a careful read. I ended up becoming frustrated and skipped most of section 5. The issues start with the abstract, where things should be kept simpler in my opinion. In particular, I struggled with the sentence L11–13, which I'm still not sure I fully understand after reading the paper. Can this be explained more simply, or perhaps omitted?

`REPLY`

`The abstract now reads: line 1-15`

`CITITATION`

`\abstract{We explore to which degree the temperature dependence of the climate radiative feedback influences the slow mode of the surface temperature response, which describes the surface air temperature adjustment to forcing on a centennial timescale. We question whether long-term climate change is described by a single e-`

folding mode with a constant timescale which is commonly assumed to be independent of temperature or forcing and the evolution of time. To do so, we analyze Atmosphere-Ocean General Circulation model (AOGCM) simulations which have an integration time of 1000 years and are forced by atmospheric $CO_2$ concentrations ranging from two times (2X) to eight times (8X) the preindustrial level. Our findings suggest that feedback temperature dependence strongly influences the equilibrium temperature response and adjustment timescale of the slow mode. The timescale of the slow mode is thus state-dependent. In addition, the effective heat capacity of the slow mode increases over time, which makes the adjustment timescale also time-dependent. The state-dependence and time-dependence of the adjustment timescale of long-term climate change call into question common eigenmode decomposition with a fast and a slow timescale, in the sense that the slow mode is not well described by a single linear e-folding mode with a constant timescale. Instead, we find that any eigenmode decomposition will depend on the forcing level, and that an additional mode or a multiple mode and timescale structure of the slow adjustment is necessary to reproduce the details of AOGCM simulated long-term climate change even at a single forcing level.
}

We changed the overall presentation of the manuscript.

COMMENT

The introduction begins rather abruptly, and assumes a fairly high level of background knowledge – for example, that it is commonly understood that the response to CO2 forcing can be decomposed into fast and slow components. The notion that climate feedbacks are temperature dependent is also assumed. I think these concepts should be introduced more slowly, with references to the relevant prior literature:

REPLY

We changed the introduction accordingly. We introduce the two-timescale approach first and mention our hypothesis. We then provide reason for our hypothesis. Finally, we illustrate the study content by a simple energy balance model.
line 15-125

CITITATION

\section{Introduction}
Studies of climate change have long found that the response of surface warming to radiative forcing occurs at multiple timescales. These studies typically differentiate between a fast timescale of response, that occurs within the first decade and is associated with the thermal inertia of the ocean's mixed layer and land, and a slow timescale of response, that occurs over centuries and is associated with the thermal inertia of the intermediate and deeper ocean \citep[e.g.][]{dickinson1998}. The latter is denoted as slow mode. Many studies have sought to understand how this slow mode of warming will unfold in time. This understanding is critical to predicting long-term warming. Commonly the slow mode has been modeled as exponential decay to equilibrium, and the decay has been assumed to be constant relative to the forcing level. In this paper we analyze the slow mode of the climate response in light of feedback temperature dependence and inconstant global heat capacity using abrupt CO$_2$ experiments with Atmosphere-Ocean General Circulation models (AOGCMs). More precisely, we analyze the timing of long-term climate change and therefore the adjustment timescale of the slow mode, which varies with temperature or forcing and time according to our study. We define the variation of the adjustment timescale with temperature or forcing as being state-dependent, and we define the variation of the adjustment timescale with the evolution of time as being time-dependent. We use this phrasing throughout the study. That is, we put forward the idea that the adjustment timescale of long-term climate change on a centennial timescale is state- and time-dependent, with the surface air temperature being the state-variable.
\newline

In this study feedback temperature dependence describes how the radiative feedback of the climate system depends on the global mean surface temperature change. The global feedback is given by the derivative of the global mean energy budget $N$ with respect to the global mean surface air temperature pertubation $T$, $\frac{d N}{d T}$. Commonly, the global feedback is assumed to be constant. However, it may change with temperature and time. We find that the dependence of the radiative feedback on temperature makes the timing of long-term climate change state-dependent. We also find a systematic time-dependent component of the adjustment timescale of the slow mode. Time-dependence of the adjustment timescale is possible due to changes in ocean heat uptake efficiency and horizontal heat transport, or an inconstant global ocean heat capacity. Oceanic timescales or heat capacities are commonly assumed to be constant. However, the circulation of the deep ocean may change \citep[e.g.][]{knutti2015} and diffusive ocean heat uptake progresses over time \citep{hansen1985,wigley1985}, which can make the deep ocean effective heat capacity inconstant. Another approach for a time-dependent timescale from a global perspective is that there exists a geographic timescale pattern that emerges from the ocean circulation or a spatial pattern of local heat capacities.
\newline

To illustrate the study content, we make use of a simple energy balance model. The simple energy balance model for the global mean surface temperature response to forcing is given by
$C \frac{\mathrm{d}T}{\mathrm{d}t}= F + \lambda T$
where $C$ is the effective heat capacity of the global system, $T$ is the surface temperature perturbation relative to a reference state, $F$ is the radiative forcing which is e.g. the forcing by atmospheric CO$_2$ or aerosols, and $\lambda$ is the feedback parameter. In equilibrium, it is called the linear forcing-feedback framework. It is a first-order differential equation which analytical solution is given by $T(t) = \frac{F}{-\lambda} (1-e^{-t/\tau})$. The equilibrium warming is

$T(\infty)=\frac{F}{-\lambda}$ and thus linear in forcing. The temporal adjustment is described by the e-folding timescale $\tau=\frac{C}{-\lambda}$ at which the single e-folding mode unfolds. That is to say, the response times of the temperature adjustment depend on $C$ and $\lambda$. The stronger the feedback parameter or the smaller the effective heat capacity, the more rapidly the system adjusts to forcing. In fact, $\lambda$ itself may depend on temperature which gives rise to temperature or forcing-dependent adjustment timescales. At the same time, $C$ may not be constant but time-dependent due to the changes in the ocean circulation and diffusive ocean heat uptake in response to global warming, or a spatial pattern of local heat capacities that prescribes the thermal inertia geographically.
\newline

…

COMMENT

What do we know about feedback temperature dependence? Is this commonly simulated by GCMs? Do we know the sign of this dependence, or is this still a subject of ongoing research? The text asserts that feedbacks become more amplifying with warming (L25), yet this is inconsistent with two out of four GCMs used in this study (Table 1). 4

REPLY

We add the following: line 81-85

CITITATION

State-of-the-art AOGCMs exhibit both positive and negative feedback temperature dependence in warming experiments under modern-day boundary conditions. There, are, however, only a few studies which quantify the degree to which the global feedback depends on temperature. \citet{bloch2015} and \citet{roe2011} have quantified feedback temperature dependence for various AOGCMs. In a recent study, \citet{bloch2020} show that feedback temperature dependence is positive for 10 out of 14 state-of-the-art GCMs.
\newline

COMMENT

Another confusing aspect for me was the introduction of the two conceptual models (Eqs. 4–6) -

What physics underlie the 1st model (based on two regions, Eqs. 4–5)? Presumably this is meant to reflect the SST pattern effect, but I don't think this was explained.

REPLY

The two-region model without the coefficient for feedback temperature dependence ($a$) mimics the pattern effect only. The pattern effect is associated with different state-variables which are the temperatures in different regions, which in turn actuate a regional radiative feedback. The pattern effect is a time-dependent radiative response and emerges from the interplay of at least two state-variables. Feedback temperature dependence introduces an additional state-dependent radiative response, since the feedback in each region now depends on temperature. In this connection, the two-region model with regional feedback temperature dependencies combines time-dependent and state-dependent feedback. We focus on the response in one effective region only. One can imagine that the surface air temperature response and radiative feedbacks are aggregated onto different regions which represent the fast mode and the slow mode. Conceptually, we analyze the slow adjustment only and therefore neglect the time-dependent radiative response associated with different state-variables, having state-dependent feedback in one region only. We add some more explanations. line 127-160

COMMENT

It would help to discuss the commonalities and differences between the two models. My understanding would be that using an efficacy term (epsilon) in the 2nd model could be mathematically equivalent to using spatially-varying feedbacks in the 1st model – is this correct? The 2nd model additionally includes a heat transport efficiency term – what physics does this involve and does it make the 2nd model different from the first?

REPLY

In the case of zero feedback temperature dependence the two-region model and the two-layer model are mathematical equivalent as demonstrated by Rohrschneider et al. (2019). They provide a thorough discussion of the two-layer model and the two-region model, which shouldn't be repeated. The efficacy factor in the two-layer model makes the fast mode and the slow mode having different radiative feedbacks. Heat transport efficiency is an inertia parameter in the two-layer model and changes the timing of climate change and the magnitude of the fast mode and the slow mode. Heat transport efficiency is associated with the two-region model by changing the heat capacity that influences the timing of climate change as well as the regional feedback parameter that gives the magnitude of the surface air temperature response. We add some more explanations. line 161-205

COMMENT

The authors ultimately choose to focus on the two-region model (Eqs. 4–5), as stated L182. Why this choice, and how does it affect the interpretation of the results? Do we even need both models in the paper? I feel like it might help to use an appendix to discuss some of the more technical aspects of the two conceptual models and/or the methodological choices, so as to keep the main text simpler and more focused on the key results and their interpretation.

REPLY

We wouldn't like to have an appendix because the conceptual models are needed to provide the theoretical background to understand the paper. We choose the two-region model because it provides a simple framework to understand the slow mode, having a simple expression for the slow mode in order to analyze the different parameters. By its nature, the two-layer model is more complicated, having more than one inertia parameter. However, it is necessary to mention the two-layer model because the slow mode can also be understood as a function of the deep ocean component, and changes in ocean heat uptake (heat uptake efficiency) changes the behavior of the slow mode. We wouldn't like to miss these conceptual insights.

The simple model section now reads: line 127-205

CITITATION

\section{Conceptual insights}
Before exploring the slow mode's behavior in AOGCMs, we provide conceptual insights about the slow mode using simple climate models. We explain conceptually the slow mode as well as demonstrate conceptually the state- and time-dependence of the adjustment timescale of the slow mode. The simple models considered here are energy balance models and outlined in detail in \citet{geoffroy2013},
\citet{geoffroy2013b}, \citet{armour2013}, \citet{rohrschneider2019}, among others.
We bring together these existing concepts to lay out the parameter dependencies of the slow mode in order to provide a solid basis and motivation for our experimental analysis. With this section we provide insight how the fast e-folding mode and the slow e-folding mode (Eq. 3) emerge from simple assumptions using energy balance models. We present two recent concepts: the two-region framework which is used in this study to analyze the slow mode; and a two-layer model in which the slow mode is a function of the Earth's deep ocean component. The two-region model is much

simpler than the two-layer model, while the two-layer model accounts explicitly for changes in the ocean circulation. However, the two-layer model can be expressed mathematically by the two-region model.
\newline

A way to represent the global mean surface temperature response to forcing is to assume two effective regions, $T=(\chi -1) T_\mathrm{F} + \chi T_\mathrm{S}$, where $\chi$ is the effective fractional area:

$$C_{\mathrm{F}} \frac{\mathrm{d} T_{\mathrm{F}}}{\mathrm{d} t}= F + (\lambda_\mathrm{F} + a_\mathrm{F} T_\mathrm{F} ) T_\mathrm{F}.$$

and

$$C_{\mathrm{S}} \frac{\mathrm{d} T_{\mathrm{S}}}{\mathrm{d} t}= F + (\lambda_\mathrm{S} + a_\mathrm{S} T_\mathrm{S} ) T_\mathrm{S}.$$

$F$ is the radiative forcing, $C$ is the constant effective heat capacity, $\lambda$ is the background feedback parameter, and $a$ is the coefficient for feedback temperature dependence. Each region behaves similarly to Eq. (2), and according to this framework, the climate response is characterized by a fast mode $T_{\mathrm{F}}$ and a slow mode $T_{\mathrm{S}}$. The two-region model without the coefficient for feedback temperature dependence ($a$) mimics the pattern effect only. The pattern effect is associated with different state-variables which are the temperatures in different regions, which in turn actuate a regional radiative feedback. The pattern effect is a time-dependent radiative response and emerges from the interplay of at least two state-variables. Feedback temperature dependence introduces an additional state-dependent radiative response, since the feedback in each region now depends on temperature. In this connection, the two-region model with regional feedback temperature dependencies combines time-dependent and state-dependent feedback. We focus on the response in the slow effective region only. One can imagine that the surface air temperature response and radiative feedbacks are aggregated onto different regions which represent the fast mode and the slow mode. Conceptually, we analyze the slow adjustment and therefore neglect the time-dependent radiative response associated with different state-variables, having state-dependent feedback in one region only. Positive feedback temperature dependence causes the equilibrium response of the slow mode to increase. Furthermore, feedback temperature dependence introduces a timescale that depends on the strength of the forcing. Considering the temporal behavior, the thermal inertia of the slow mode is represented by a single effective heat capacity which is much higher than the heat capacity of the fast mode ( $C_{\mathrm{F}} \ll C_{\mathrm{S}}$). At this point, $C_{\mathrm{S}}$ is constant over time and does not change with the climate state. .
\newline

Another conceptual framework with a fast mode $T_{\mathrm{F}}$ and a slow mode $T_{\mathrm{S}}$ is the two-layer ocean model with ocean heat uptake efficacy and feedback temperature dependence \citep{held2010,winton2010}. We extend this model by introducing a coefficient for feedback temperature dependence. This model then also combines time-dependent feedback due to the evolution of two different state-variables and state-dependent feedback due to temperature-dependent feedback. The model configuration with ocean heat uptake efficacy and feedback temperature dependence is given by

$$C\frac{\mathrm{d} T}{\mathrm{d} t}= F + (\lambda_{\mathrm{b}} + a T) T - \epsilon \eta (T-T_{\mathrm{D}})$$

$$C_{\mathrm{D}} \frac{\mathrm{d} T_{\mathrm{D}}}{\mathrm{d} t}= \eta (T-T_{\mathrm{D}})$$

where $C \ll C_{\mathrm{D}}$ are the heat capacities of the upper- and deep-ocean, $\lambda_{\mathrm{b}}$ is the background feedback parameter and $a$ the coefficient for feedback temperature dependence. The parameter $\eta$ is the heat transport efficiency and $\epsilon$ the efficacy factor for ocean heat uptake. The slow component is approximated by

$$T_\mathrm{s} (t) \approx \frac{\sqrt{\Lambda^2 - 4 a F}-{\sqrt{\Lambda^2- 4 a F - 4 a \epsilon \eta T_\mathrm{D} (t)}}}{2 a} \: \: \text{with} \: \: \Lambda=\lambda_\mathrm{b}- \epsilon \eta$$

after the fast contribution from the surface, as derived in \citet{rohrschneider2019}. Following this conceptual framework, the slow mode is a function of the deep ocean component $T_\mathrm{D}$ because the slow mode emerges from the heat transport into the deep ocean and the convergence of the state-variables over time towards the same equilibrium temperature perturbation.
\newline

Using linear model versions without feedback temperature dependence, the two-region model and the two-layer model are mathematically equivalent. There is no difference in the fast e-folding mode and the slow e-folding mode (Eq. 3) as well as in the global radiative response between these models. Although no analytical solution of the coupled two-layer model with feedback temperature dependence exists to date, we can approximate the temperature and radiative response associated with the slow mode by a single effective region (Eq. 5), having a single heat capacity. However, the parameters of the two-layer model modify the inertia of the slow mode. For instance, the parameter for the efficiency of ocean heat uptake $\eta$ is an inertia parameter, and changes in ocean heat uptake cause $C_{\mathrm{S}}$ to increase or decrease. Commonly, we assume that the parameters which describe these simple models are constant. In that respect, we emphasize that the slow mode's response is described by

$$C_{\mathrm{S}} \frac{\mathrm{d} T_{\mathrm{S}}}{\mathrm{d} t}=N_\mathrm{S}$$

where $N_\mathrm{S}$ is the TOA imbalance associated with the slow mode. After having explored the imprint of feedback temperature dependence on the slow mode, we analyze the interplay of state-varying and time-varying adjustment timescales. The former arises from the presence of feedback temperature dependence while the latter arises from the inconstancy of $C_{\mathrm{S}}$ according to Eq. (5,9).
\newline

For the experimental analysis, we choose the two-region model because it provides a simple framework to understand the slow mode, allowing us to analyze the different parameters. By its nature, the two-layer model is more complicated, having more than one inertia parameter and an efficacy term. However, it is necessary to mention the two-layer model because the slow mode can also be understood as a function of the deep ocean component, and changes in ocean heat uptake or in the heat uptake efficiency do change the behavior of the slow mode. Heat uptake efficiency is an inertia parameter in the two-layer model and it changes the timing of climate change and the magnitude of the fast mode and the slow mode while the magnitude of the global equilibrium response remains unchanged. Heat transport efficiency is associated with the two-region model by changing the heat capacity of the region that influences the timing of climate change as well as the regional feedback parameter that gives the magnitude of the surface air temperature response. In the following we analyze briefly the parameter dependencies of the equilibrium response and timescale of the slow mode using the two-layer ocean model. We use the more complicated two-layer ocean model to show the dependence of

```
the slow mode on the ocean circulation besides feedback temperature dependence. We
focus on the heat uptake efficiency in the two-layer model to provide an
explanation for changes in C$_S$. It, C$_S$, may change with model parameter, and
this model parameter may also vary with time. It is not straightforward to find a
simple analytical expression for the dependence of C$_S$ on $\eta$.
\newline
```

```
...
```

COMMENT

I would like the authors to clarify and make explicit their definition of temperature-dependent feedbacks. It seems to me that there are two quite distinct types of temperature dependence: (a) a temperature-dependent SST pattern effect, versus (b) temperature-dependent feedback processes (independent of the SST pattern). The latter could be quantified for example using uniform SST warming or cooling experiments. My understanding is that the temperature dependence discussed in the present paper includes both processes (a) and (b), but it would be good to clarify this. Do the authors know which type of temperature dependence is more important for their findings? If we want to understand and perhaps observationally constrain the temperature dependence of climate feedbacks, it seems to me that different approaches would be needed for (a) versus (b).

```
REPLY
```

```
We do not focus on the pattern effect because we
analyze the response of one effective region only
onto which the surface air temperature response and
radiative  feedback  of  the  slow  adjustment  is
aggregated. Feedback temperature dependence is the
second-order  temperature  dependence  of  the  first-
order  radiative  feedback,  using  Taylor-series. We
hope  it  is  clear  now  with  the  changes  made  in  the
introduction  and  the  section  on  the  conceptual
models. We look forward to have your opinion.
```

SPECIFIC COMMENTS

We meet the specific comments listed below.

L24: "As a result" – of what?
L112: Should clarify that this isn't the formulation used by Held et al. and Winton et al. (who didn't consider feedback temperature-dependence, as far as I'm aware?) Agreed
L185: Shouldn't it be N_F(t=0)?
L186–188: I wasn't able to follow this, can you explain in more detail or illustrate this graphically? (After further reading, I see this is explained more clearly L225–227. This needs to be reorganised.)
L193–196: Again I wasn't able to fully follow. I'd recommend explaining this in more detail in an appendix.
L248: remove extra "between"
L283–284: I didn't follow this reasoning.
Answerd
L441: "publicly *available* experiments"
L443: The reference to year 2100 is odd, considering that the results are based on idealised step forcing experiments, rather than realistic RCP-style scenarios. (On this timescale the slow mode arises, and it behaves like in the case of step  function input)